

# Spatio-temporal variation characteristics of global wildfires and their emissions

Hao Fan[1], Xingchuan Yang[2], Chuanfeng Zhao[3], Yikun Yang[3], Zhenyao Shen[1]

[1]State Key Joint Laboratory of Environmental Simulation and Pollution Control, School of Environment, Beijing Normal University, Beijing, 100875, China

[2]College of Resource Environment and Tourism, Capital Normal University, Beijing 100048, China

[3]Laboratory for Climate and Ocean-Atmospheric Studies, Department of Atmospheric and Oceanic Sciences, School of Physics, Peking University, Beijing 100871, China

*Correspondence to: Chuanfeng Zhao (cfzhao@pku.edu.cn)*

**Abstract.** Intense regional wildfires are a common occurrence in the context of climate warming and have progressively evolved into one of the major natural disasters in terrestrial ecosystems, posing a serious hazard to the atmosphere and climate change. We investigated the spatial distribution, intensity, emission changes, and meteorological differences of wildfires in different wildfire active and wildfire-prone regions globally based on multi-source satellite remote sensing fire data, emission data, and meteorological data in order to better understand the change trend of wildfire activity at multiple spatial and temporal scales. The findings demonstrate that while the wildfire burned area (BA) has decreased slowly over the last 20 years, the wildfire burned fraction (BF), the fire count (FC), and the fire radiative power (FRP) all exhibit pronounced regional and seasonal variations. The physical characteristics of wildfires, including the BF, FC and FRP, experience greater seasonal variation as latitude increases, with summer and autumn as the reasons with the most frequent wildfires worldwide. This study also shows that the emission declined substantially between 2012 and 2020 in Northern Canada, Alaska, and Northeast China, whereas it notably increased in the Siberia region during the same period, primarily due to a rise in summer emissions. The results based on classification show that the absolute amount of $CO_2$ produced by wildfires is the largest, and the difference among regions is relatively small. Excluding $CO_2$, aerosol emissions (the total of OC, TC, and BC) ranged from 78.6% to 84.2%, while the least significant air pollutants (the total of $PM_{2.5}$, $SO_2$, and NOx) ranged from 5.8% to 11.7%. The abundance of vegetation predominantly affects the intensity change of wildfire development,



while the weather conditions can also indirectly influence the incidence of wildfire by altering the growth

condition of vegetation. Correspondingly, the increase in temperature in the northern hemisphere's middle and high latitude forest regions was primarily responsible for the increase in wildfires and emissions, while the change in wildfires in tropical regions was more influenced by the decrease in precipitation and relative humidity. This study contributes to the understanding of regional variations in wildfire activity and emission variability, and provides support for the control of wildfire activity across

regions and seasons.



## 1  Introduction

A significant natural disturbance factor that can directly damage the surface vegetation and pose a serious threat to biodiversity is fire (Akagi et al., 2011; Requia et al., 2021; Turetsky et al., 2015). A lot of aerosols,

greenhouse gases, and trace gases are released into the atmosphere during wildfires, which has an impact on the chemistry of the atmosphere and the carbon cycle in ecosystems (Bian et al., 2007; Fan et al., 2021; Jaffe et al., 2004; Permar et al., 2021; Turetsky et al., 2015). More studies are identifying wildfire as a significant contributor to global climate change as a result of the numerous negative effects it has on the atmospheric environment (Ding et al., 2021; Kaulfus et al., 2017; Liang et al., 2022).

Although wildfires have an impact on climate change, there is evidence that climate has an even greater impact on global biomass burning than human activities in some periods (Marlon et al., 2013). In addition to directly influencing the likelihood of wildfires, rainfall and temperature can indirectly modify vegetation productivity (i.e., fuel richness), which in turn influence the intensity of wildfires (Engelmann et al., 2021; Kloss et al., 2019; Yu and Ginoux, 2022). Rainfall before the growing season is frequently a

significant barrier to fire activity in areas with limited biomass, according to earlier literature (Govender et al., 2006; Jolly et al., 2015). However, in regions with a lot of biomass, the main causes of wildfire activity are high temperatures and seasonal drought (Bowman et al., 2020; Requia et al., 2021). Extreme and prolonged droughts caused by abnormal changes in the weather-climate system, such as El Nino, have long been a key driver of regional wildfires (Andela and van der Werf, 2014; Kloss et al., 2019). In

addition to natural variables, human activities also have an impact on the occurrence and progression of wildfires. Land management and fire control strategies are among these activities, and they are also significantly influenced by population density, socioeconomic development, and surface landscape (Bistinas et al., 2014; van der Werf et al., 2006; Zheng et al., 2021).

Wildfires have a harmful effect on the climate and human society. On one hand, wildfires can

substantially worsen air quality and endanger human health by spewing out harmful gases. For instance, while anthropogenic emissions of air pollutants in China considerably fell during the COVID-19 pandemic in early 2020, contaminants from wildfires on the Indochina Peninsula caused abnormal rises in $PM_{2.5}$ and CO concentrations in southwest China (Fan et al., 2021). Similar to this, large wildfires in southeast Australia in 2019 led to an increase in aerosol optical density and carbonaceous aerosols of

over 30% (Ohneiser et al., 2022; Yang et al., 2021). Additionally, research has revealed that wildfire-





produced aerosol particles raise the risk of premature death by an estimated 5-8% globally, particularly in tropical locations where flames are more likely to endanger human life and property (Requia et al., 2021). Wildfires, on the other hand, create aerosols that can have an impact on local, regional, and even global radiation balances. For instance, the transport of Russian biomass-burning aerosols over South

Korea reduces sun radiation by 57% (Lee et al., 2005).

In fact, a series of studies have been carried out on short-term wildfire events at the regional scale based on station observations, aircraft measurements and model simulations (Bian et al., 2017; Giglio et al., 2010; Lu et al., 2016; Yu and Ginoux, 2022). Researchers have also proposed indicators like Fire Count (FC) and Burned Area (BA) to describe the physical properties and processes of wildfires in order to

better explain wildfire characteristics and discriminate different types of wildfires (Senande-Rivera et al., 2022; Zheng et al., 2021). The quantitative descriptions of wildfire events in various regions, however, are still extremely variable according to the findings of previous studies (Andela et al., 2017; Liang et al., 2022), necessitating further consideration of both global and regional factors as well as in-depth research into the reasons behind regional variations. The majority of the world's regions, particularly

those with forests in mid- and high-latitudes, will see a future with a higher danger of wildfires as global warming progresses (Yu and Ginoux, 2022; Zhu et al., 2021). Clearly, it is not enough just to target key regions with typical wildfire events, such as those in the western United States, Australia, and central Africa (Damoah et al., 2004; Xue et al., 2021; Yu et al., 2021). Meanwhile, the information regarding the emissions of various compounds caused by wildfires is still debatable at this time, and the model

simulation research has a larger inaccuracy than the observation inversion data (Zhang et al., 2016; Zheng et al., 2021). Therefore, it is necessary to strengthen long-term, systematic investigations of wildfire frequency, intensity, and emission characteristics on a worldwide basis.

Based on multi-source remote sensing data and the Global Fire Assimilation System (GFAS), this study will systematically analyze the long-term changes in wildfire distribution, intensity, and emissions at

global and regional scales. At the same time, this study will quantify the regional variations in wildfire characteristics, and discover the importance of meteorological factors influencing the wildfire behavior. The study intends to: (1) quantify the patterns of worldwide distribution of wildfire count, area, and intensity; (2) examine the characteristics and regional variations of aerosols and greenhouse gases generated by wildfires; and (3) investigate the causes by integrating straightforward climatic indicators.



This study adds to our understanding of worldwide and typical long-term trends in wildfire activity and

multi-substance emissions, and supports future predictions of wildfire and emission changes in various

places. This paper is organized as follows: The primary summary is presented in Section IV after Section

III gives the results and discussions and Section II outlines the data and methods used in this study.

## 2 Data and methods

### 2.1 Study area

To better analyze the variation characteristics and regional differences of wildfires, this study focused on

two spatial scales: the global scale and 12 typical regions with frequent wildfires. The selection of 12

regions considered the vegetation growth and cover, the distribution and intensity of wildfire activity

over the years, and the combination of typical areas from previous case studies (Spracklen et al., 2015;

Yang et al., 2021; Zheng et al., 2021). The names and latitude-longitude ranges of the 12 typical regions

are shown in Table 1, with the specific spatial distribution to be further explained in Section III.

### 2.2 Fire, Emission and Weather Data

### 2.2.1 Fire data

The MODIS sensors on the Terra and Aqua platforms are capable of monitoring fires and are currently

an important source of global data on fire locations and areas burned (Giglio et al., 2009; Yang et al.,

2021). MCD64 maps the daily burned area globally at a spatial resolution of 500 m by combining MODIS

surface reflectance, fire activity, and vegetation cover information. MCD64 uses active fire observations

to analyze the statistical characteristics of burning-related and non-burning-related changes, and uses

Bayesian probability testing methods to classify burned or unburned grid cells across the globe (Giglio

et al., 2018). In this paper, monthly MCD64 Collection 6 products with a spatial resolution of 0.25°×0.25°

from 2001 to 2019, namely MCD64CMQ products, are used.

The MOD14 product based on Terra sensor and the MYD14 product based on Aqua sensor are the most

widely used fire products, and they have a spatial resolution of 375 m. The MODIS satellite is fully

automated in fire detection and can produce daily global fire information, including geographic

coordinates of fire, date and time of fire occurrence, and confidence coefficient, etc. This paper uses fire

count and fire radiative power of the global MCD14DL data from 2001 to 2019 with the confidence



coefficient more than 50%.

FireCCI51 is the latest version of the global burning area developed under the FireCCI project (Lizundia-Loiola et al., 2020). It is an improved version of FireCCI50 and generates the world's first BA product with 250 m spatial resolution. FireCCI51 combines MODIS spectral information at 250 m spatial resolution with thermal anomaly information from MODIS active fire products to produce global burned area products.

### 2.2.2 Fire Emission data

The wildfire emission data used in this study is the Global Fire Assimilation System (GFAS), which has been widely used in previous studies (Fan et al., 2021; Kaiser et al., 2012; Pan et al., 2020). The NASA's Terra and Aqua MODIS active fire products are used by the GFAS to estimate daily fire emissions with a horizontal resolution of 0.1° (Kaiser et al., 2012).

### 2.2.3 Climate data

ERA-5 is the fifth generation of atmospheric reanalysis meteorological data developed by the European Centre for Mesoscale Weather Forecasts, providing many atmospheric variables with a horizontal resolution of 0.25° × 0.25°, and a time resolution of up to 1 hour. Compared to ERA-Interim, ERA-5 provides substantial improvements in both horizontal and vertical resolution, covering the period since 1979 (Albergel et al., 2018). The ERA-5 reanalysis data ingests more data sources, uses an updated numerical weather prediction model and data assimilation system, and is widely used (Fan et al., 2021; Guo et al., 2021; Yang et al., 2021). In this study, the 2 m temperature, total precipitation, relative humidity, and soil moisture from the ERA-5 dataset are used. Table 2 shows the specific names, resolutions, and period information of the above multi-source observation data and reanalysis data.

### 2.3 Analysis method

The statistical methods used in this study are all conventional statistical methods, which are specified as follows. In this paper, the spatial-temporal distribution characteristics of global fire activity and intensity are analyzed by combining burned area (BA), burned fraction (BF), fire count (FC), and fire radiative power (FRP). The BA and BF are from the MCD64CMQ dataset, where BF is the proportion of burned area to the total area in each grid cell. The FC and FRP are point data with geographic location



information from the MCD14DL dataset. In this paper, fire points are distributed in a global grid of 0.25°×0.25° according to their latitude and longitude to obtain the average value of each grid at yearly and seasonal time scales.

Wildfire season is involved in the influence of meteorological elements on wildfire. Here, the specific extraction method of wildfire season is to rank the monthly burned area in each natural year. The month with 80% of the annual average burned area is the fire month, and the fire month number is the duration of the wildfire season. The advantage of this classification is that it is completely independent of burned area and does not make any assumptions about seasonal patterns of burning. For example, it can accommodate bimodal fire patterns during the year (i.e., there are two fire seasons in a year). In addition, the calculation of coefficient of variation (CV) is also used in this study to characterize and analyze the uncertainty of various emissions. The CV was calculated as the standard deviation of the data divided by the mean.

We use geographic detector to quantify the contribution of meteorological conditions (temperature, relative humidity, soil moisture, and total precipitation) to wildfire changes in different regions. The geographic detector can explain the degree of variability of various independent variables (x) to dependent variable (y). The q statistic in the calculation results indicates the degree of interpretation of the corresponding variable and its value range is 0-1. The larger the q is, the stronger the explanatory power of (x) to (y) is. The geographic detector model has currently been used extensively in research for quantitative attribution analysis. Detailed description of this model can refer to the studies by Wang et al. (2010, 2016).

## 3   Results and Discussion

### 3.1. Global temporal and spatial variation characteristics of wildfires

The BA, which is frequently employed in existing studies as a single metric to quantify wildfire changes (Senande-Rivera et al., 2022; Zheng et al., 2021), shows a steady decreasing trend in both observed data and model simulation results (Fig. 1). This study employs the yearly average spatial distribution of BF, FC, and FRP from 2001 to 2019 to thoroughly depict the spatial pattern of wildfire occurrence and evolution in order to examine and evaluate the distribution characteristics of global wildfire in a more systematic manner (Fig. 2). In general, BF and FC represent the fire area, whereas FRP is more indicative



of fire intensity. Figures 2a and 2b reveal that the spatial pattern of BF presented by the MCD64CMQ and FireCCI51 datasets is essentially the same, both demonstrating great spatial heterogeneity.

From the perspective of spatial distribution (Fig. 2), the tropics, particularly in Africa, northern Australia, and central South America, are regions where the high values of BF are mostly found. Specifically, sub-Saharan Africa and northern Australia have BF values that generally surpass 20%, and the greatest value even reaches 40%. At the same time, boreal forest distribution zones in Canada, western United States, central Europe, and Russia constitute the banded high wildfire occurrence zones in the middle and high latitudes of the Northern Hemisphere. In addition, parts of Southeast Asia and southern North America also have high fire rates, with BF averages of about 10-20%. In contrast, the prevalence of fire is low, with BF below 10%, in southeast China, South Asia, southeast of the United States, and southeast of South America. It is evident that the arid and semi-arid regions with less vegetation cover (e.g., the western China, the Middle East, the Sahara Desert) and polar regions with lower temperatures have extremely low fire rates. This shows that the vegetation abundance (i.e., availability of fuel) is a key factor in determining the likelihood of wildfires (Huangfu et al., 2021; Liu et al., 2017; Turetsky et al., 2015).

The overall spatial distribution for FC is similar to that for BF (Fig. 2c). As FC reflects more fire point quantity information, it shows sporadic and spotty high value distribution of boreal coniferous forest, and contiguous distribution of fire points in tropical rain forest and grassland. The multi-year average FC in southern Africa exceeds 50, and the average FC in the western United States, Alaska and the boreal forest of Russia is about 30-50. In the arid and semi-arid areas with sparse vegetation cover and areas with low temperature, there is almost no fire point. This further reflects that the number and size of wildfire burning points are spatially heterogeneous globally.

FRP in Fig.2d shows the global wildfire burning intensity. The boreal forest regions in southwestern Australia, North America, and Russia have high FRP average values that can surpass 120 MW, although BF in these areas is not high. In contrast, FRP released during fires were not high in regions where fires occurred most frequently, such as the Amazon Basin, African savannas and rainforests, and northern Australia, which is consistent with previous findings (Chen et al., 2018; Kumar et al., 2022; Ohneiser et al., 2022; Requia et al., 2021; Yu et al., 2021). This is mostly due to two factors: vegetation types and climatic conditions. Compared to savanna areas, temperate forest areas have more biomass and more



combustible tree species, such as spruce. However, compared to herbaceous fires, wildfires in wooded environments spread more slowly, and canopy fire generation is minimal in savannas and woods. On the other hand, frequent rainfall in tropical areas compared to temperate wildfire burn zones may reduce fire intensity, particularly when temperate forest areas undergo prolonged drought, leading to much higher

fire intensity (Jaffe et al., 2004; Konovalov et al., 2021; Zhuang et al., 2021). It should be noted that the FRP data integrates all the radiation energy from the 1 km$^2$ window, thus it includes both radiation from open flame burning and radiation from smoldering (non-open flame combustion). In actual, fires in grassland areas provide the majority of the radiation power that has been observed, whereas smolder is more common in forested areas in both cold and temperate zones. Since BF does not effectively show

the occurrence of such smoldering, there is a certain difference between FRP and BF. It is necessary to examine the distribution of global and regional wildfire physical characteristics from the perspective of a multi-index combination in light of the aforementioned combustion kinds and data identification disparities.

The seasonal average spatial distribution characteristics of BF, FC, and FRP worldwide from 2001 to

2019 are depicted in detail in Fig. 3. In terms of seasonal distribution, the global fire activity in spring is relatively low, and there are large BF, FC and FRP values only in southern Africa, northern Southeast Asia, and northern Australia. BF, FC, and FRP reach annual peaks in the rainforests of southern Africa, Central Asia, and the temperate and boreal forest regions of the Northern Hemisphere during summer, when there is a major rise in wildfire activity worldwide. Autumn fire activity increased a lot in northern

Australia, south-central South America, the African savanna, and the Malay Islands, while it declined in central Asia and the southern African rainforests. The most noticeable change during the winter is a steep decline in fire activity in the northern hemisphere, with strong fire activity primarily happening in the western and northern parts of Australia and the African savanna. In general, there are obvious spatial differences in the area, quantity and intensity of global wildfires. The seasonal variation of wildfires

increases, especially as latitude increases. This is mostly due to the fact that appropriate meteorological conditions and biomass fuels work together to control wildfire occurrence (Kloss et al., 2019; Zhang et al., 2022; Zhuang et al., 2021). Additionally, although there were only minor seasonal variations in the physical characteristics (BF, FC, and FRP) defining wildfires, there were relatively obvious regional variations (Figs. 2 and 3).




The northern part of the Indochina Peninsula, and the northeastern part of China are experiencing an increase in fire activity, which is primarily related to agricultural activities like burning crop residue in these regions. This is especially true with the intensification of agriculture and the expansion of agricultural scale, which may result in an increase in burning under human control (Andela et al., 2017; Feng et al., 2021). In contrast, wildfire activity is reducing in Central Asia, which is correlated with a

wetter environment and an increase in grazing lands (Hao et al., 2021). Because agricultural land is still being abandoned widely and there are fewer fires started by seasonal burning, the eastern Europe and western Russia saw a sharp decrease in fire activity (Jaffe et al., 2004; Konovalov et al., 2021). In the Oceania area, while fire activity in Australia's northern section generally decreased, it increased in its western and southeast regions, which may be tied to ENSO occurrences (Andela et al., 2014; Yang et al.,

2021; Yu et al., 2021).

### 3.2 Inter-regional wildfire variation characteristics and emission difference analysis

Based on the analysis in Section 3.1, we find that the distribution and intensity of global wildfire have spatial heterogeneity. According to the spatial distribution characteristics of BF, FC, FRP and Normalized Difference Vegetation Index (NDVI), and combined with previous studies, 12 fire-prone regions were

divided to further analyze the physical variation characteristics and emission differences of wildfire among regions (Table 1). Fig. S1 shows the average annual spatial distribution of global NDVI from 2000 to 2019 and the geographical locations of typical regions in this study.

We first examine the relative changes in BA, FRP, and plume top height (APT) among the 12 wildfire regions. The analysis period is slightly adjusted to 2003-2020 because we used GFAS data for this portion.

In 12 wildfire regions, the relationship between APT, FRP, and BA was nonlinear and lacked regular regional variation features, as shown in Fig. 4. The relative change difference of APT is less than that of BA and FRP, and the differences among the 12 regions are basically between ±20%, and the relative change difference of BA is concentrated between ±80%. Regarding the FRP, the differences among other regions except NCA are relatively consistent and mainly distributed within the range of ±60%. APT, like

FRP, can not only be used as an index to measure wildfire intensity, but also reflects the ability of wildfire emissions to affect the environment and climate to a certain extent. This is because the higher the wildfire plume, the greater the range of environmental pollution and climate forcing effects are likely (Bian et al., 2017; Hennigan et al., 2012; Konovalov et al., 2021).





The relative changes of BA, APT, and FRP of NAU are all the highest, as observed in Fig. 4, suggesting

that the wildfire occurred most frequently with high intensity. This conclusion is in line with recent

research showing an increase in mega-wildfire incidents and a rise in the regional transport contribution

of biomass burning aerosols in Australia (Ohneiser et al., 2022; Yang et al., 2021). SI, WUS and NCA

are characterized by low BA but high FRP, which strongly confirms that the intensity of wildfires in

forest areas of middle and high latitudes in the northern hemisphere is high, so the probability of wildfires

in temperate and boreal forests may increase under the background of warming climate. The areas with

positive relative changes in BA and close to 0 or even negative relative changes in APT and FRP are

mainly concentrated in tropical area, which reflects that tropical area has a larger burning range for a

long time period, but the wildfire intensity and plume height are not prominent.

How the wildfire emissions change with time is an important topic that needs to be studied urgently at

present, because it can better reflect its environmental and climate impact potential than simply

describing the changes of wildfire physical characteristics. However, the existing studies, especially the

simulation quantification of wildfire emission by the models, is uncertain and even has an opposite trend

(Pan et al., 2020; Zheng et al., 2021). For example, studies have shown that the decline in global fire

emissions is partly due to reduced deforestation in tropical area of Asia, including Indonesia, resulting in

lower fire emissions (Feng et al., 2021; van der Werf et al., 2010). On the contrary, a new study shows

that despite the global decline in wildfires, the trend of increasing forest burning has not proportionately

reduced global carbon emissions from fires (Zheng et al., 2021). Here, we examine and compare emission

characteristics and differences across 12 regions and several periods using GFAS emission flux data, one

of the currently recognized mainstream wildfire emission data (Figs. 5 and 6). The gas CO is frequently

treated as a crucial tracer for wildfires and their emissions (Bian et al., 2007; Ding et al., 2021; Hooghiem

et al., 2020). To compare and assess the emission trend throughout the two time periods of 2003 to 2011

and 2012 to 2020, we started the study by using CO emission flux as the core index (Fig. 5). The findings

demonstrate that the emissions from wildfires exhibit distinctive regional and seasonal features. In

contrast to SI and NCA, which have the most pronounced seasonal fluctuations and higher absolute

values of the emission flux, only EAMZ, SAF, and NAF exhibit relatively mild seasonal changes.

Comparing the two periods of 2003-2011 and 2012-2020, most of the 12 regions have similar seasonal

characteristics of emissions, and the absolute value of emission flux is also close. The change of global



wildfire emission flux in recent years is not consistent, with most areas showing a decreasing trend or basic invariability. NCA and NEC showed a significant decrease in the mean value of emission fluxes from 2012 to 2020, while SI showed a significant increase in the mean value of emission fluxes from 2012 to 2020. Combining with Fig. 4, we can conclude that while the FRP of NCA and SI are both high, the variations in the emission flux over the last 20 years are totally at odds with one another. Therefore, from the perspective of emissions, the future emission potential of SI region is large, and the increase of emissions is mainly concentrated in summer, which is closely related to the temperature increase in Siberia and the whole Northern Hemisphere high latitude region in recent years, especially the increase of hot weather in spring and summer (Evangeliou et al., 2018; Rantanen et al., 2022). In fact, WUS emissions also increased in summer while decreased in winter in recent years, so the multi-year average emissions have not changed significantly (Fig. 5).

In order to more systematically reveal the variation characteristics of wildfire emissions and the proportion of different emitted substances in each region, $CO_2$, $CH_4$ and $N_2O$ fluxes were selected to characterize greenhouse gases (GHG) emissions; $PM_{2.5}$, $SO_2$ and $NO_x$ fluxes were used to characterize air pollutant emissions; OC, TC and BC fluxes were used to characterize aerosol emissions (Fig. 6). The cumulative emission fluxes of NCA and SI were the highest in all regions, while those of SAF and NAF were the lowest. The cumulative emission fluxes of wildfire generally showed spatial distribution characteristics that increased with the rise of latitude. The emission difference among regions is not only reflected in the absolute value of emission flux, but also has obvious seasonal characteristics. Specifically, NCA, WUS and WAMZ showed unimodal distribution of high emissions in summer, CE and NEC showed unimodal distribution of high emissions in winter, NAU and NSEA showed unimodal distribution of high emissions in autumn and spring, respectively, while SI showed relatively large variation with multiple peaks and valleys in spring, summer and autumn. Except for the above regions, the intra-year variation is relatively flat and the seasonal differences are small. The seasons with the highest wildfire occurrence and emissions were mainly summer and autumn at a worldwide scale (Figs. 3 and 6).

From the perspective of the proportion of each substance, the highest emission flux is $CO_2$ (accounting for more than 90%), which is far greater than the sum of the emission fluxes of other substances, and shows good emission consistency among the study regions. If $CO_2$ is removed, the aerosol emission flux



(the sum of OC, TC and BC) accounts for 78.6%-84.2%, followed by air pollutants (the sum of $PM_{2.5}$, $SO_2$ and $NO_x$) that account for 5.8%-11.7%, and the sum of $CH_4$ and $N_2O$ accounts for the least, which is about 3.2%-7.3%. In this way, there are differences in the emissions of different types of substances

among the study regions, and greenhouse gas emissions dominated by $CO_2$ are absolutely dominant. The $CO_2$ released by wildfires is bound to affect the global carbon cycle and has a feedback effect on the climate system. On one hand, climate warming may increase the occurrence of wildfires, which release more $CO_2$ into the atmosphere, thereby exacerbating the global warming and forming a positive feedback. On the other hand, due to the impact of land use change (such as urban expansion, deforestation, and

land reclamation), the BA has shown a downward trend in the past 20 years (Fig. 1). The reduced wildfire will, to some extent, slow down the trend of global warming by increasing the terrestrial carbon sink, forming a negative feedback process (Wu et al., 2021, 2022). Despite a general decrease in the area subject to wildfires, a recent study finds that global emissions of carbon from wildfires have not decreased (Zheng et al., 2021). Therefore, carbon emissions from wildfires will continue to be an

important source of global carbon cycle in the future, and the warming effect brought by wildfires cannot be ignored. Not only that, the environmental problems caused by air pollutants and aerosols from wildfires are equally serious. For example, the particulate matter and CO pollutants emitted by wildfires in the spring of 2020 caused widespread air pollution in Southeast Asia and East Asia (Fan et al., 2021), and multi-year wildfire aerosol emissions in Southeast Asia significantly enhanced low cloud generation

over the northern South China Sea (Ding et al., 2021).

Obviously, absolute value differences of wildfire emission flux are visible, but its stability and uncertainty remain unknown. To reflect the degree of dispersion of each type of emission data in this study, we specifically calculated the yearly CV value of each type of emission flux (Fig. 7). First, the mean CV for all emission fluxes was over 150%, with the minimum of 120% and the maximum over

300%. This fully indicates that the emissions generated by wildfire vary greatly. Combined with the results in Figs. 5 and 6, we can preliminarily judge that the emission uncertainties found in this study mainly come from regional differences and seasonal variation characteristics. Secondly, it can be found that although the CV values of emission fluxes of all substances are large, the dispersion degree of GHG emissions is the lowest (161), air pollutants are the highest (202), and aerosol (197) is in between. Thus,





this study confirms that compared with aerosol and air pollutant, wildfires produce more greenhouse

gases with the smallest regional difference, especially $CO_2$ (Figs. 6 and 7).

**3.3. Meteorological driving factors in wildfire burned areas**

Based on the findings of earlier research, it can be found that biomass fuel is a necessary condition for

wildfire activity, and meteorological conditions are an important influencing factor (Andela et al., 2014;

Zhuang et al., 2021). Existing studies tend to analyze wildfire activity and its driving factors in a certain

region, or use fire weather index and other comprehensive meteorological indices to establish a

correlation with wildfire events (Grillakis et al., 2022). However, these results are difficult to answer how

the meteorological factors affecting wildfire activity in different regions. To explore the influence of

meteorological conditions on wildfire activities in different regions, this study selected four indicators of

temperature, precipitation, relative humidity, and soil moisture for analysis.

Fig. 8 shows the global annual trends of temperature, precipitation, relative humidity, and soil moisture

from 2001 to 2019. As can be seen from Fig. 8a, the temperature in most parts of the global land shows

an upward trend, especially in parts of the Northern hemisphere at high latitudes with a larger increase

(>0.05°C/ year). On the contrary, the temperature in the northern United States, central and eastern

Canada, northern Central Asia, and northern India showed a downward trend, most of which were about

-0.05-0°C/ year. The variation of global total precipitation is relatively complex. The increase and

decrease of precipitation over the ocean are significantly larger than that over the land, and its distribution

is more consistent with the influence of atmospheric circulation (Fig. 8b). In terms of the land areas of

interest, the precipitation in eastern North America, Northern Europe, South Asia and eastern East Asia

showed an increasing trend, while the precipitation in western North America, central and eastern Europe,

Siberia, Indochina Peninsula and North China showed a decreasing trend. In contrast, precipitation

changes are larger in the tropics and the southern hemisphere, with obvious polarization. For example,

in northwestern South America, Africa and east Malaysia, precipitation increased obviously (> 0.025

mm/year), while precipitation in eastern and southern South America, central and southern Africa, and

northern Australia showed a decreasing trend (about -0.020 mm/year). Both relative humidity and soil

moisture can have a direct impact on the occurrence of wildfire and can also have an indirect impact by

affecting the transpiration and development of vegetation (Tian et al., 2022; Yue et al., 2017). The relative

humidity showed an obvious upward trend in central North and South America, South Asia and the



Tibetan Plateau region, while it mainly decreased in other continental regions (Fig. 8c). The change of
soil moisture is relatively fragmented, because it is more easily disturbed by human activities (such as
irrigation, farming, etc.) than other variables. Note that the change of soil moisture is also affected by
precipitation (Figs. 8b and 8d).

This study further investigated the variation trend and spatial distribution characteristics of temperature,
precipitation, relative humidity and soil moisture during the wildfire season (Fig. S2). In general, the
variation trend of meteorological factors in wildfire season is similar to the annual variation trend in
spatial pattern. Due to the large regional and seasonal differences in meteorological factors, this study
conducted annual and seasonal sliding trend analysis of each variable in each region to further explore
the influence of meteorological factors in each region on fire activities (Figs. S3-S6). In 2001-2009, the
precipitation in western Amazon decreased in summer, and the relative humidity and soil moisture
showed a declining trend in summer and autumn, which were favorable for the occurrence and spread of
fire. As a result, the peak value of BA in the western Amazon rose prior to 2007 before falling, which
was congruent with the rise in relative humidity and soil moisture. BA in the eastern Amazon showed a
slight increasing trend, which may be related to the increase of temperature, the decrease of precipitation
and soil moisture in the past decade. From the perspective of meteorological factors, the significant
increase of relative humidity and soil moisture can promote the growth of vegetation, which is conducive
to vegetation recovery and reduces the risk of wildfire (Brandt et al., 2017; Yue et al., 2017). In recent
years, the precipitation in central Europe has increased, and the relative humidity and soil humidity have
increased after decreasing, which is conducive to the growth of vegetation and makes the air more humid.
At the same time, the reduction of summer temperature also reduces the risk of wildfires (Chen et al.,
2020; Hao et al., 2021).

The temperature of NCA increased significantly in recent years (Fig. 8), and precipitation (summer and
autumn), relative humidity (spring and summer) and soil moisture (spring and winter) showed an
increasing trend. As a result, the plant growth period was prolonged in these locations as precipitation
and soil moisture increased, which was beneficial to the growth of vegetation and hence more
combustible materials were stored. However, higher temperatures in summer increased the risk of
wildfire occurrence, which eventually caused increased emissions from wildfires (Junghenn Noyes et al.,
2022). Similar fluctuations and reasons are present in WUS wildfire activity (Kaulfus et al., 2017; Lu et



al., 2016; Xue et al., 2021). In recent years, SI has revealed a considerable rise in wildfire emissions, particularly during the summer (Fig. 5). This is mainly due to the increase of temperature in SI area

(especially in spring and summer), the significant decrease of precipitation and soil moisture (summer and autumn), and the non-significant downward trend of relative humidity. In addition, the rise of temperature and the loss of soil moisture in winter further increased the possibility of wildfire disasters in winter, which is conducive to the formation of smoldering phenomenon in SI (Chan et al., 2020; Konovalov et al., 2021). The above results indicate that, on the one hand, climate warming has greatly

improved vegetation productivity and increased biomass fuels, especially in the extratropical Northern Hemisphere (Zhang et al., 2022). At the same time, a warmer climate promotes snow melt, which causes wildfire seasons to start earlier but end later. On the other hand, global warming leads to higher summer temperatures and increased drought conditions, thus increasing the risk of extreme fires.

Finally, we quantified the impact of meteorological factors on wildfire changes in different regions based

on the statistical model of geographical detectors (Table 3). Globally, the influence of temperature and relative humidity on wildfires is relatively high, which can explain 24% and 33% of the causes of wildfires, while the contribution of total precipitation is low and not significant. Regionally, the changes of wildfires in the middle and high latitudes of the Northern Hemisphere are mainly affected by temperature. The proportion of temperature that can explain the change of wildfires in NCA, WUS, NCE,

CE, and SI areas is 32%, 42%, 38%, 42%, and 19%, respectively. These areas are long in winter and covered with snow, with few open fires. Wildfire activities and high emission periods are mainly concentrated in summer and autumn, so the response of middle and high latitude wildfire changes to temperature changes is more sensitive. In contrast, wildfire activities in tropical areas (including Amazon, Africa, Southeast Asia and northern Australia) are more sensitive to relative humidity, soil moisture and

precipitation. This may be because the temperature change in different seasons in these areas is not obvious, but the drought caused by water vapor reduction is more likely to promote the increase of wildfires (Bowman et al., 2020; Brandt et al., 2017).

As shown in Table 4, the ability of interactions between meteorological factors to explain wildfire changes has been greatly improved compared with single meteorological factor. Especially for Amazon

(WAMZ, EAMZ) and Africa (NAF, SAF), where human activities are relatively less involved, natural meteorological elements can explain more than 70% of wildfire changes. In areas with more human



activities, human control has greatly changed the fire, and thus the contribution from interpretations of meteorological factors is relatively limited, which generally does not exceed 60%. For example, in the eastern United States, Western Europe, and eastern regions in China, despite the change of meteorological

factors is large, fire activity has, in the past 20 years, generally been stable, which is mainly attributed to the fact that most of the land surface types are urban land, with less combustible substances, and the surface landscape (such as road network) has divided the vegetation area (Andela et al., 2017; Fan et al., 2021; Feng et al., 2021; Zhang et al., 2018). In reality, there are many natural and human factors that can cause and affect wildfires, such as lightning, arson, etc., which have not been considered in this study.

This research here mainly attempts to conduct quantitative exploration from the perspective of atmospheric environment that may affect wildfire changes.

## 4   Summary and Conclusions

Multi-source satellite remote sensing data (MCD64CMQ, MCD14DL, and FireCCI51) were used in this study to analyze the spatiotemporal characteristics of global fire activity and intensity at time scales over

the last 20 years, and wildfire emission data (GFAS) were used to quantitatively analyze emission fluxes and seasonal variations in typical regions. Finally, using the ERA-5 meteorological data, it assessed the effects of fire and fire-duration meteorological factors on wildfire burning and emissions at the global and typical regional levels. This study clarifies the global and regional wildfire activities and their emission characteristics, and provides a reference for quantifying the impact of meteorological factors

on wildfires. Specific conclusions are as follows:

1. The global wildfire burning area showed a slow decreasing trend. The wildfire activities (BF, FC and FRP) were quite different among regions, and had seasonal variation characteristics of high in summer and autumn, and low in winter and spring. Particularly, the seasonal fluctuation of wildfire events increased with the increase of latitude. Fuel richness and climate conditions are the key factors that

determine the occurrence and development of wildfire.

2. Although the wildfire burned area has decreased over the past 20 years, there hasn't been a corresponding reduction in the amount of emissions caused by fires worldwide. The primary type of emissions from wildfires is greenhouse gases with $CO_2$ dominant, followed by aerosol and air pollution. There were differences in emissions among the 12 typical regions. The tropical region and the Southern


Hemisphere showed relatively weak changes in emissions, while the increased wildfire activity in

forested regions of middle and high latitudes in the northern hemisphere, especially in summer, was the

main cause of the overall increase in regional emissions in the past decade.

3. Regional variations in meteorological conditions clearly affect the frequency and severity of wildfires.

When there is less human interference, variations in climatic variables including temperature,

precipitation, relative humidity, and soil moisture have a stronger correlation with changes in fire activity

and intensity. For instance, meteorological factors can explain more than 70% of wildfire changes in the

Amazon and African rainforests. In regions with relatively strict artificial fire management, such as the

Eastern United States, Western Europe and China, wildfire activity has generally been stable in the past

20 years. This is mostly attributable to the development of artificial fire suppression techniques and the

division of vegetative regions into surface landscapes and built-up urban areas.

**Data Availability.**

The    MOD14    product    and    FireCCI51    dataset    can    be    downloaded    from

https://firms.modaps.eosdis.nasa.gov/    (last    access:    27    November    2022),    and

https://climate.esa.int/en/projects/fire/data/ (last access: 2 November 2022) respectively. The FireCCI51

dataset can be downloaded from MCD64CMQ from the University of Maryland, Department of

Geographical Sciences (sftp://fuoco.geog.umd.edu, last access: 2 November 2022). Wildfire emission

data from the Global Fire Assimilation System (GFAS) (https://apps.ecmwf.int/datasets/data/cams-gfas/,

last access: 30 November 2022). ERA-5 Reanalysis data were provided by the European Centre for

Medium Weather Forecasts, (https://cds.climate.copernicus.eu/, last access: 2 November 2022).

**Acknowledgements**

This work was supported by the National Natural Science Foundation of China (grants
41925022,42205178), the China Postdoctoral Science Foundation (2022M720459).

**Author contributions.**

CFZ designed the research, and CFZ and HF carried out the research and wrote the manuscript. XCY

provided constructive comments and revised the manuscript many times. YKY and ZYS provided

constructive comments on this research. All authors made substantial contributions to this work.





**Competing interests.**

The authors declare that they have no conflict of interest.

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



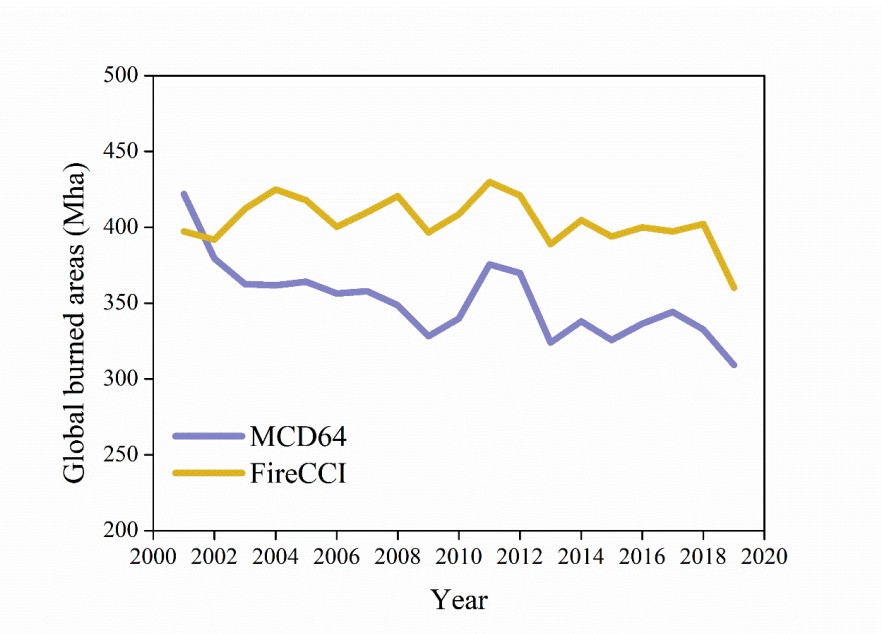

**Figure 1. Global burned areas derived from MCD14 and FireCCI.**

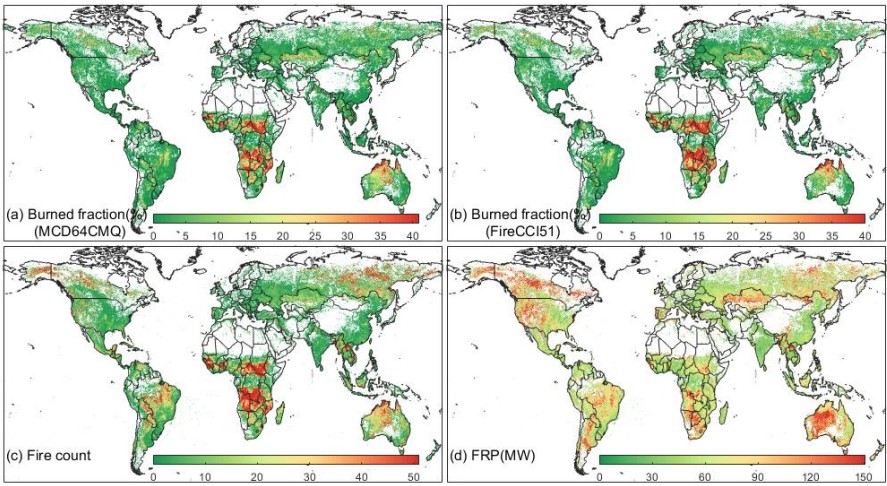

**Figure 2. The spatial distribution of global fire burned fraction (BF), fire count (FC) and fire radiation power (FRP) from 2001 to 2019.**



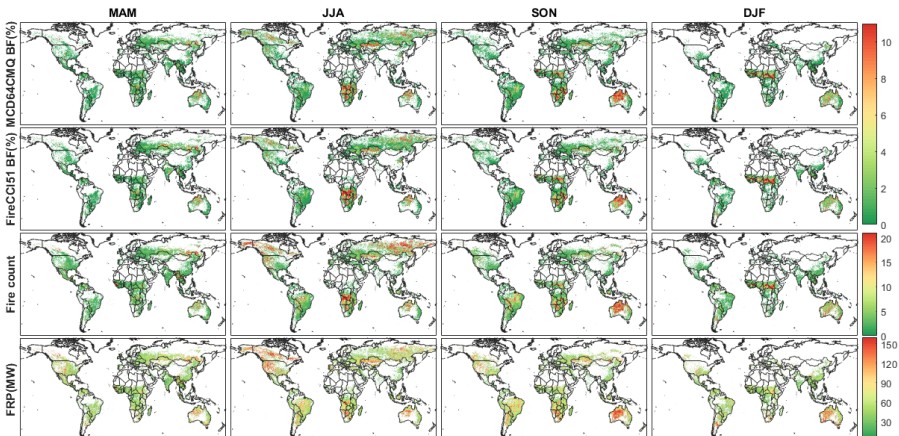

**Figure 3. Seasonal spatial distribution of global fire burned fraction (BF), fire count (FC) and fire radiation power (FRP) from 2001 to 2019.**

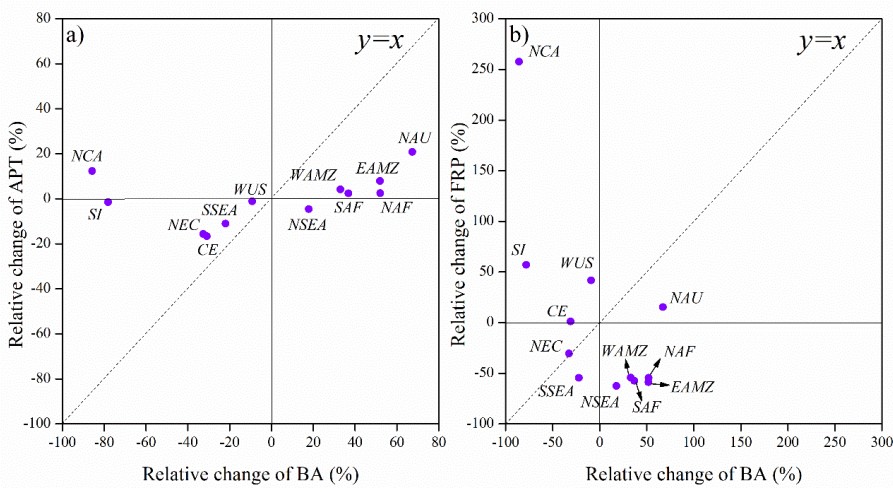


**Figure 4. The relative changes of the burning area (BA), plume top height (APT) and fire radiation power (FRP) of wildfires in 12 regions.**

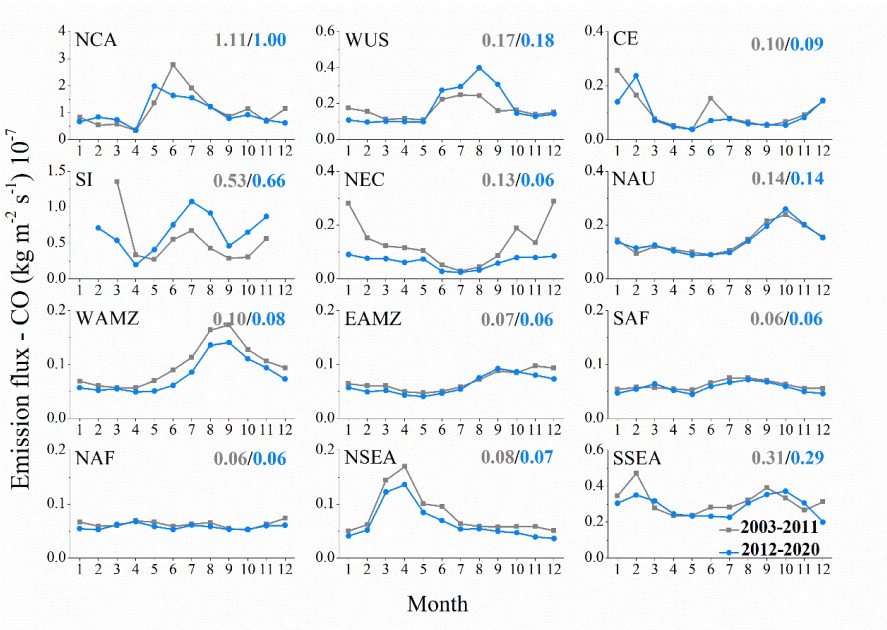

**Figure 5. Comparison of changes in CO emissions between the two periods. The values in the figure represent the average emission flux of the two periods respectively.**

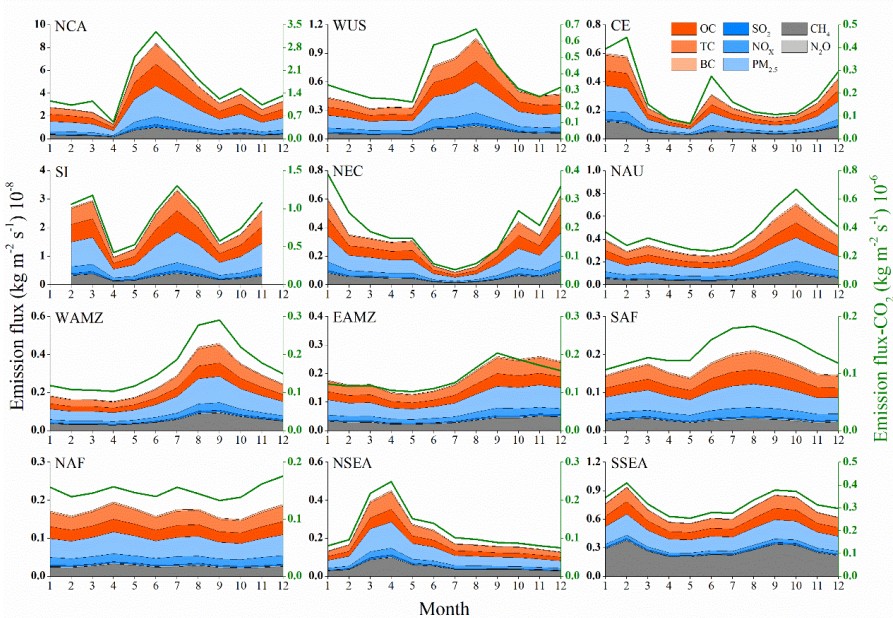

**Figure 6. Cumulative distribution of emissions in 12 regions. The right coordinate axis is used for CO2, and the left coordinate axis is used for other emission.**





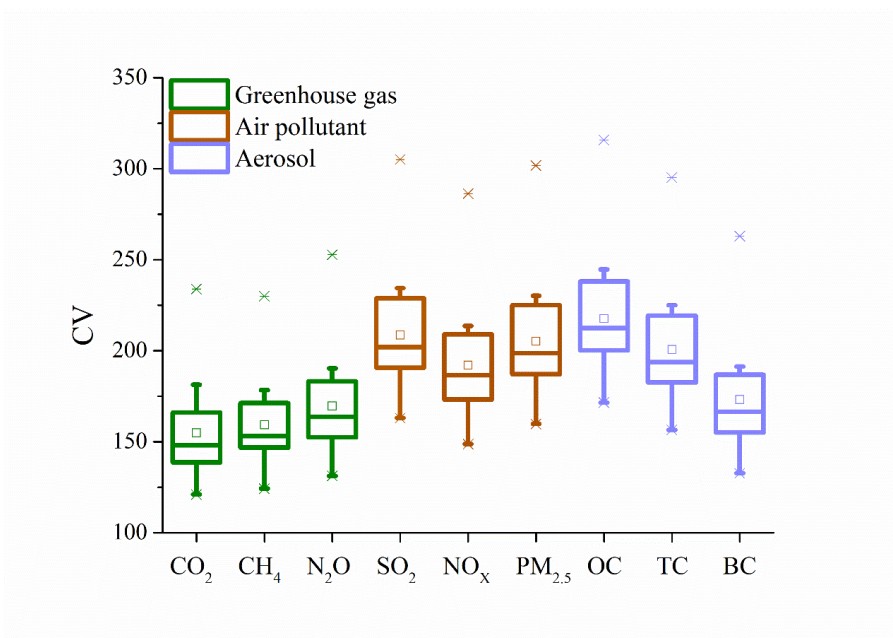


**Figure 7. The variation coefficient (CV) of regional emissions.**

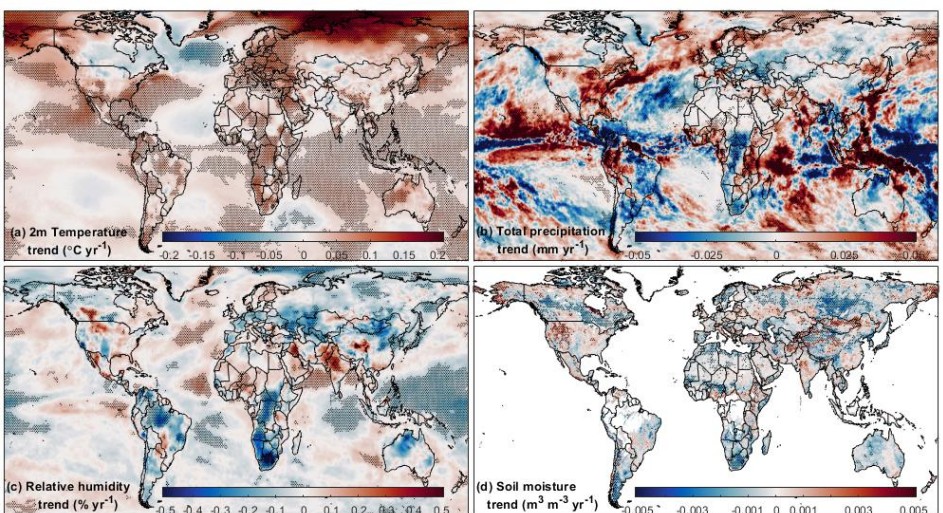

**Figure 8. Global trends in temperature, total precipitation, relative humidity (1000 hpa) and soil moisture from 2001 to 2019. The "*" in the figure represents that the trend has passed the 95% significance test.**




**Table 1. 12 regions and their geographical locations.**

| Region | latitude / Longitude |
|---|---|
| Northern Canada and Alaska (NCA) | 55°N-70°N,165°W-105°W |
| Western United States (WUS) | 30°N-49°N,125°W-100°W |
| Northeast China (NEC) | 40°N-54°N,122°E-135°E |
| Northern Australia (NAU) | 11°S-23°S,120°E-150°E |
| Siberian Area (SI) | 55°N-70°N,90°E-140°E |
| Western Amazon (WAMZ) | 22°S-0°,70°W-50°W |
| Eastern Amazon (EAMZ) | 22°S-0°,50°W-35°W |
| Northern Africa (NAF) | 5°N-15°N,10°W-35°E |
| Southern Africa (SAF) | 16°S-2°N,10°E-35°E |
| Central European (CE) | 45°N-55°N,30°E-75°E |
| North of Southeast Asia (NSEA) | 6°N-25°N,91°E-110°E |
| South of Southeast Asia (SSEA) | 10°S-6°N,95°E-150°E |

**Table 2. Summary of datasets used in this study.**

| Product | Dataset name | Resolution | Period |
|---|---|---|---|
| MCD64CMQ | Burned Area | Monthly,0. 25°×0.25° | 2001.1-2019.12 |
| MCD14DL | Fire count, Fire radiative power | Daily, point | 2001.1-2019.12 |
| FireCCI51 | Burned Area | Monthly,0. 25°×0.25° | 2001.1-2019.12 |
| GFAS | Emissions | Monthly,0. 1°×0.1° | 2003.1-2020.12 |
| ERA-5 | 2m Temperature | | |
| | Relative humidity | | |
| | Soil moisture | Monthly,0.25°×0.25° | 2001.1-2020.12 |
| | Total precipitation | | |






**Table 3. Influence degree of meteorological factors on wildfire changes in different regions. T is temperature, RH is relative humidity, SM is soil moisture and P is total precipitation.**

| | | T | RH | SM | P |
|---|---|---|---|---|---|
| Global | q statistic | 0.24 | 0.33 | 0.20 | 0.04 |
| | p value | 0.00 | 0.00 | 0.01 | 0.82 |
| NCA | q statistic | 0.32 | 0.19 | 0.16 | 0.11 |
| | p value | 0.00 | 0.00 | 0.00 | 0.11 |
| WUS | q statistic | 0.42 | 0.33 | 0.32 | 0.24 |
| | p value | 0.00 | 0.00 | 0.00 | 0.00 |
| NEC | q statistic | 0.38 | 0.27 | 0.06 | 0.16 |
| | p value | 0.00 | 0.00 | 0.20 | 0.00 |
| SI | q statistic | 0.19 | 0.14 | 0.20 | 0.12 |
| | p value | 0.00 | 0.00 | 0.00 | 0.05 |
| WAMZ | q statistic | 0.23 | 0.58 | 0.78 | 0.36 |
| | p value | 0.00 | 0.00 | 0.00 | 0.00 |
| EAMZ | q statistic | 0.08 | 0.83 | 0.71 | 0.54 |
| | p value | 0.10 | 0.00 | 0.00 | 0.00 |
| NAF | q statistic | 0.36 | 0.38 | 0.24 | 0.52 |
| | p value | 0.00 | 0.00 | 0.00 | 0.00 |
| SAF | q statistic | 0.64 | 0.90 | 0.77 | 0.84 |
| | p value | 0.00 | 0.00 | 0.00 | 0.00 |
| CE | q statistic | 0.42 | 0.43 | 0.37 | 0.05 |
| | p value | 0.00 | 0.00 | 0.00 | 0.54 |
| NSEA | q statistic | 0.44 | 0.63 | 0.59 | 0.68 |
| | p value | 0.00 | 0.00 | 0.00 | 0.00 |
| SSEA | q statistic | 0.08 | 0.19 | 0.30 | 0.23 |
| | p value | 0.14 | 0.01 | 0.08 | 0.00 |
| NAU | q statistic | 0.14 | 0.55 | 0.56 | 0.52 |
| | p value | 0.01 | 0.00 | 0.00 | 0.00 |




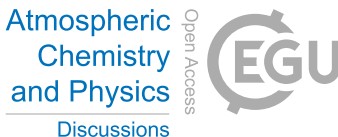

**Table 4. The impact of interaction between meteorological factors on wildfire changes in different regions. T is temperature, RH is relative humidity, SM is soil moisture and P is total precipitation. The bold number indicates that the interaction factor has a significant difference in the impact on wildfire compared with a single meteorological factor.**

| | | T∩RH | T∩SM | T∩P | RH∩SM | RH∩P | SM∩P |
|---|---|---|---|---|---|---|---|
| Global | q statistic | 0.74[b] | 0.57[b] | **0.36[b]** | **0.58[b]** | **0.62[b]** | **0.45[b]** |
| NCA | q statistic | 0.40[a] | **0.36[a]** | **0.42[a]** | 0.38[b] | 0.45[b] | 0.27[a] |
| WUS | q statistic | 0.52[a] | 0.50[a] | **0.55[a]** | 0.44[a] | 0.51[a] | 0.45[a] |
| NEC | q statistic | 0.55[a] | **0.53[b]** | 0.51[a] | **0.47[b]** | 0.32[a] | 0.41[b] |
| SI | q statistic | 0.39[b] | 0.37[a] | 0.25[a] | 0.38[b] | 0.38[b] | 0.46[b] |
| WAMZ | q statistic | **0.84[b]** | **0.90[a]** | **0.75[b]** | **0.85[a]** | **0.73[a]** | **0.81[a]** |
| EAMZ | q statistic | **0.90[a]** | **0.83[b]** | **0.69[b]** | **0.84[a]** | **0.84[a]** | **0.74[a]** |
| NAF | q statistic | 0.95[b] | 0.93[b] | **0.89[b]** | **0.76[b]** | 0.63[a] | **0.79[b]** |
| SAF | q statistic | **0.95[a]** | **0.94[a]** | **0.89[a]** | **0.92[a]** | **0.94[a]** | **0.95[a]** |
| CE | q statistic | 0.55[a] | 0.56[a] | **0.63[b]** | 0.55[a] | **0.58[b]** | **0.57[b]** |
| NSEA | q statistic | **0.81[a]** | **0.82[a]** | **0.76[a]** | 0.70[a] | 0.73[a] | **0.73[a]** |
| SSEA | q statistic | 0.38[b] | **0.46[b]** | **0.39[b]** | 0.44[a] | 0.44[b] | 0.42[a] |
| NAU | q statistic | **0.72[b]** | **0.72[b]** | **0.63[a]** | 0.64[a] | 0.58[a] | 0.62[a] |

[a] indicates that the interaction belongs to bi-factor enhancement.

[b] indicates that the interaction belongs to nonlinear enhancement.