# Peer review of "Spatio-temporal variation characteristics of global fires and their emissions"

_Atmospheric Chemistry and Physics, 2022_

## Referee Comment (RC1)

The paper "Spatio-temporal variation characteristics of global wildfires and their emissions" presents a very detailed analysis of fire burned area, fire fraction, fire radiative power, and fire count in different regions and different seasons using satellite remote sensing data, emission data, and meteorological data. The study found that while the burned area of wildfires has decreased slightly over the past 20 years, there are pronounced regional and seasonal variations in the burned fraction, fire count, and fire radiative power of wildfires. The study also found that emissions from wildfires decreased in some regions (such as Northern Canada, Alaska, and Northeast China) while increasing in others (such as Siberia) and that the intensity of wildfire development is primarily affected by the abundance of vegetation, while weather conditions can also indirectly influence wildfire incidence. The study concludes by suggesting that this research can provide support for the control of wildfire activity across regions and seasons.

The paper is well-structured, the research question is well-defined, and the methods used are appropriate for the research question. The study findings are well presented and the conclusion is well drawn.

General comments:
1. Introduction: it would be better to include more papers from previous studies in your paper, particularly those focused on burned area trends, fire intensity trends, global patterns of wildfires, variations in fire season, and the impact of weather on wildfires. These studies would provide a more comprehensive understanding of the topic.

2. This paper found a decrease in burned areas in the western US, however, other studies have found an increase in burned areas in the same region. For example:

   M. Burke, A. Driscoll, S. Heft-Neal, J. Xue, J. Burney, M. Wara, The changing risk and burden of wildfire in the United States. Proc. Natl. Acad. Sci. U.S.A. 118, e2011048118 (2021).

   Y. Zhuang, R. Fu, B. D. Santer, R. E. Dickinson, A. Hall, Quantifying contributions of natural variability and anthropogenic forcings on increased fire weather risk over the western United States. Proc. Natl. Acad. Sci. U.S.A. 118, e2111875118 (2021).

   This discrepancy in findings is noteworthy and deserves further investigation. I would recommend considering comparing the findings in your paper with previous studies and discussing the differences.

3. This study utilized GFAS emission data, however, it has been shown in the paper by Li et al. (2020) that GFAS data is much lower than other widely used emission data products. This difference in the data sets could have a significant impact on the results of your study. I recommend comparing GFAS with other emission data products (e.g., QFED, FEER, GFED, etc.) to make your results more robust and reliable.
   Li, Y., Tong, D. Q., Ngan, F., Cohen, M. D., Stein, A. F., Kondragunta, S., et al. (2020). Ensemble PM2.5 forecasting during the 2018 Camp Fire event using the HYSPLIT transport and dispersion model. Journal of Geophysical Research: Atmospheres, 125, e2020JD032768. https://doi.org/10.1029/2020JD032768

Specific comments:

1. Line 59: Please add more references for wildfire impact on human health. Below are some examples:

   Reid, C.E., Brauer, M., Johnston, F.H., Jerrett, M., Balmes, J.R., Elliott, C.T.: Critical review of health impacts of wildfire smoke exposure. Environ. Health Perspect. 124, 1334–1343, 2016.

   Cascio WE.: Wildland fire smoke and human health. Sci Total Environ. doi: 10.1016/j.scitotenv.2017.12.086, 2018.

   O'Neill, S. M., Diao, M., Raffuse, S., Al-Hamdan, M., Barik, M., Jia, Y., Reid, S., et al.: A multi-analysis approach for estimating regional health impacts from the 2017 Northern California wildfires, Journal of the Air & Waste Management Association, 71:7, 791-814, DOI: 10.1080/10962247.2021.1891994, 2021.

   Liu, Y., Austin, E., Xiang, J., Gould, T., Larson, T., & Seto, E.: Health impact assessment of the 2020 Washington State wildfire smoke episode: Excess health burden attributable to increased PM2.5 exposures and potential exposure reductions. GeoHealth, 5, e2020GH000359. https://doi. org/10.1029/2020GH000359, 2021.

   Li, Y., Tong, D., Ma, S., Zhang, X., Kondragunta, S., Li, F., & Saylor, R.: Dominance of wildfires impact on air quality exceedances during the 2020 record-breaking wildfire season in the United States. Geophysical Research Letters, 48, e2021GL094908. https://doi.org/10.1029/2021GL094908, 2021.

2. Introduction: Please include more papers on burned area trends, fire intensity trends, global patterns of wildfires, variations in fire season, and the impact of weather on wildfires.

3. Line 109: Please provide the full name for MODIS. Make sure you have provided the full name of any abbreviation before using it for the first time in your paper. Please check throughout the paper.

4. Line 110: full names for MCD, MCD64CMQ, and MCD14DL are missing.

5. Line 117: full names for MOD, and MYD are missing.

6. Line 117: Please add the reference for the FIRMS product here.

7. Line 123: full name for FireCCI is missing.

8. Section 2.2.2: Compare GFAS with other widely used fire emission products. (See general comments)

9. Line 134: full names for ERA, ERA-Interim, and ERA5 are missing.

10. Line 153: any reference for the definition of the fire month? Why do you use 80% rather than 70% or another percentage?

11. Figure 1: It would be better to show the trend of burned areas in the 12 different regions defined in table 1.

12. Figure 1: Please explain the difference in the results of MODIS and FireCCI.

13. Figure 2: Please add the time period in the figure 2 caption.

14. Line 199: It's interesting to see how land use affects the fire trend. It'd be better to add one figure and some discussion about the BA, BF, BC, and FRP trends in different land use type regions.

15. Line 254: According to Sofiev et al. (2012), plume injection height is related to FRP. It'd be better to show the scatter plot of FRP and APT.

    Sofiev, M., Ermakova, T., & Vankevich, R.: Evaluation of the smoke-injection height from wild-land fires using remote-sensing data. Atmospheric Chemistry and Physics, 12(4), 1995–2006. https://doi.org/10.5194/acp-12-1995-2012, 2012.

16. Line 264: The relative changes of FRP of NAU is not the highest. NCA, SI, and WUS is higher than NAU.

17. Line 274-303: This paragraph is very long. Consider separating it into two paragraphs.

18. Figure 5: Can you explain why there is no data for SI in the Jan and Feb (2003-2011)?

19. Line 295: The increase in SI is only after May. There is a decrease from Jan-April.

20. Figure 6: Please provide the time period in the caption.

21. Line 388: why use 2001-2009 instead of 20 years (2001-2019)?

---

## Author Comment (AC1)

We thank the reviewer for his/her thoughtful, valuable and detailed comments and suggestions that have helped us improve the paper quality. Our detailed responses (Blue) to the reviewer's questions and comments (*Italic*) are listed below.

*Reviewer #1:*

*The paper "Spatio-temporal variation characteristics of global wildfires and their emissions" presents a very detailed analysis of fire burned area, fire fraction, fire radiative power, and fire count in different regions and different seasons using satellite remote sensing data, emission data, and meteorological data. The study found that while the burned area of wildfires has decreased slightly over the past 20 years, there are pronounced regional and seasonal variations in the burned fraction, fire count, and fire radiative power of wildfires. The study also found that emissions from wildfires decreased in some regions (such as Northern Canada, Alaska, and Northeast China) while increasing in others (such as Siberia) and that the intensity of wildfire development is primarily affected by the abundance of vegetation, while weather conditions can also indirectly influence wildfire incidence. The study concludes by suggesting that this research can provide support for the control of wildfire activity across regions and seasons.*
*The paper is well-structured, the research question is well-defined, and the methods used are appropriate for the research question. The study findings are well presented and the conclusion is well drawn.*

We highly appreciate the positive comments from the reviewer which encourage us a lot. We also thank the reviewer for the detailed suggestions and comments. We have gone through all the comments and revised the original manuscript accordingly.

*General comments:*
*1. Introduction: it would be better to include more papers from previous studies in your paper, particularly those focused on burned area trends, fire intensity trends, global patterns of wildfires, variations in fire season, and the impact of weather on wildfires. These studies would provide a more comprehensive understanding of the topic.*

We agree with the reviewer and appreciate the suggestion. We have added the introduction at Lines 77-91: "**In terms of BA, forest fires made up the majority of the area burned in Equatorial Asia, followed by the North America. Savanna fires were extremely prevalent in Africa and considerably less in South America. Farmland fires were the most prevalent in Europe and the Middle East, while grassland fires were dominant in Central Asia and South America (Giglio et al., 2013; van Wees et al., 2022). The majority of the world's regions, particularly those with forests in mid- and high-latitudes, will see a future with a higher danger of wildfires as global warming progresses (Yu and Ginoux, 2022; Zhu et al., 2021).**

**In the past decade or so, although the reduction of man-made fires in tropical areas has led to the reduction of the global area of over-fire, the trend in other regions is on the rise, the frequency of extreme wildfire events is increasing, and the difference in seasonal variation is more obvious (Bowman et al., 2020; Senande-Rivera et al., 2022; Zheng et al., 2021). Several recent studies (Huang et**

**al., 2023; Xu et al., 2022) have found that the BA of wildfires in the West Bank of the United States and the Indo-China Peninsula in Southeast Asia has increased and has significant synoptic scale changes, and the strongest frequency spectrum is in the time scale of 1 week and 2 weeks, respectively. The former is controlled by wind speed and humidity, while the latter is mainly modulated by rainfall (Huang et al., 2023).**".

*2. This paper found a decrease in burned areas in the western US, however, other studies have found an increase in burned areas in the same region. For example:*
*M. Burke, A. Driscoll, S. Heft-Neal, J. Xue, J. Burney, M. Wara, The changing risk and burden of wildfire in the United States. Proc. Natl. Acad. Sci. U.S.A. 118, e2011048118 (2021).*
*Y. Zhuang, R. Fu, B. D. Santer, R. E. Dickinson, A. Hall, Quantifying contributions of natural variability and anthropogenic forcings on increased fire weather risk over the western United States. Proc. Natl. Acad. Sci. U.S.A. 118, e2111875118 (2021).*
*This discrepancy in findings is noteworthy and deserves further investigation. I would recommend considering comparing the findings in your paper with previous studies and discussing the differences.*

Sorry for the confusion. In this study, we found that the burning area of fires in the western United States has been rising slowly in the past 20 years, and this overall conclusion is basically consistent with the above two studies. However, there are some differences between our study and these two studies, as shown in Table R1. In addition, our newly added Fig. 4 also directly shows the increasing trend of BA in the western US.

Following the reviewer's suggestion, we added relevant discussion about the comparison at Lines 196-200: "**The BA, which is frequently employed in existing studies as a single metric to quantify fire changes (Senande-Rivera et al., 2022; Zheng et al., 2021), shows a steady decreasing trend based on both global observation data and model simulations (Fig. 1). However, there is also an increasing trend of BA in many specific regions, such as the Arctic and the western United States (Burke et al., 2021; Engelmann et al., 2021; Zhuang et al., 2021)**.".

Table R1. Comparison of differences among the three studies.

|  | Study area | The period |
|---|---|---|
| Burke et al., PNAS | Public and private US lands | 1983-2020 |
| Zhuang et al., PNAS | The Western US | 1979-2020 (May to September) |
| This study | The Western US | 2001-2020 |

*3. This study utilized GFAS emission data, however, it has been shown in the paper by Li et al. (2020) that GFAS data is much lower than other widely used emission data products. This difference in the data sets could have a significant impact on the results of your study. I recommend comparing GFAS with other emission data products (e.g., QFED, FEER, GFED, etc.) to make your results more robust and reliable.*
*Li, Y., Tong, D. Q., Ngan, F., Cohen, M. D., Stein, A. F., Kondragunta, S., et al. (2020). Ensemble PM2.5 forecasting during the 2018 Camp Fire event using the HYSPLIT transport and dispersion*

*model. Journal of Geophysical Research: Atmospheres, 125, e2020JD032768. https://doi.org/10.1029/2020JD032768*

We appreciate the useful information and comment from the reviewer.

As the reviewer said, GFAS, QFED, FEER, GFED and FINN are widely used emission data, but there are also some differences among them. The main factors causing the difference of the above emission data include fire detection, emission factor, biome types and burning stages. The reason why we chose the GFAS data was formally carefully screened, and we referred to the paper on the comparison of these emission data (Pan et al., 2020). Relevant studies (Kaiser et al., 2012; Pan et al., 2020) found that the biomass burning OC emissions derived from GFED3.1, GFED4s, FINN1.5, GFAS1.2, FEER1.0, and QFED2.4 can differ by up to a factor of 3.8 on an annual average, with values of 15.65, 13.76, 19.48, 18.22, 28.48, and 51.93 Tg C, respectively. The biomass burning BC emissions can differ by up to a factor of 3.4 on an annual average.

In general, higher biomass burning emissions are estimated from QFED2.4 globally and regionally, followed by FEER1.0. GFAS data ranked in the middle of the global and regional emissions assessment, showing relatively stable performance and effectively avoiding overestimation and underestimation in regional studies (Kaiser et al., 2012; Pan et al., 2020). More importantly, the object of this study is to analyze the changes of global and typical regional wildfire emission trends and the attribution of differences in emission change trends, rather than absolute differences. Therefore, we finally used GFAS data after comprehensive consideration. We modified the description of GFAS data at Lines 141-149: "**The fire emission data used in this study is the Global Fire Assimilation System (GFAS), which has been widely used in previous studies (Fan et al., 2021; Kaiser et al., 2012; Li et a., 2020; Pan et al., 2020). Note that different products could provide biomass burning OC emissions with large differences (Kaiser et al., 2012; Pan et al., 2020), introducing uncertainties to our analysis. For example, Li et al. (2020) suggested the low biases of GFAS in biomass emissions. However, there are also studies (Kaiser et al., 2012; Pan et al., 2020) showing that GFAS is more stable than other data in the description of fire emissions, which is suitable for the analysis and comparison of fire emission trends in this study. Thus, the NASA's Terra and Aqua MODIS active fire products are used by the GFAS to estimate daily fire emissions with a horizontal resolution of 0.1° (Kaiser et al., 2012).**".

Kaiser, J. W., Heil, A., Andreae, M. O., Benedetti, A., Chubarova, N., Jones, L., Morcrette, J.-J., Razinger, M., Schultz, M. G., Suttie, M., and van der Werf, G. R.: Biomass burning emissions estimated with a global fire assimilation system based on observed fire radiative power, Biogeosciences, 9, 527–554, https://doi.org/10.5194/bg-9-527-2012, 2012.

Pan, X., Ichoku, C., Chin, M., Bian, H., Darmenov, A., Colarco, P., Ellison, L., Kucsera, T., da Silva, A., Wang, J., Oda, T., and Cui, G.: Six global biomass burning emission datasets: intercomparison and application in one global aerosol model, Atmos. Chem. Phys., 20, 969–994, https://doi.org/10.5194/acp-20-969-2020, 2020.

Li, Y., Tong, D. Q., Ngan, F., Cohen, M. D., Stein, A. F., Kondragunta, S., Zhang, X., Ichoku, C.,

Hyer, E. J., and Kahn, R. A.: Ensemble $PM_{2.5}$ forecasting during the 2018 camp fire event using the HYSPLIT transport and dispersion model, J. Geophys. Res.-Atmos., 125, e2020JD032768, https://doi.org/10.1029/2020JD032768, 2020.

*Specific comments:*

1. *Line 59: Please add more references for wildfire impact on human health. Below are some examples:*

   *Reid, C.E., Brauer, M., Johnston, F.H., Jerrett, M., Balmes, J.R., Elliott, C.T.: Critical review of health impacts of wildfire smoke exposure. Environ. Health Perspect. 124, 1334–1343, 2016.*

   *Cascio WE.: Wildland fire smoke and human health. Sci Total Environ. doi: 10.1016/j.scitotenv.2017.12.086, 2018.*

   *O'Neill, S. M., Diao, M., Raffuse, S., Al-Hamdan, M., Barik, M., Jia, Y., Reid, S., et al.: A multi-analysis approach for estimating regional health impacts from the 2017 Northern California wildfires, Journal of the Air & Waste Management Association, 71:7, 791- 814, DOI: 10.1080/10962247.2021.1891994, 2021.*

   *Liu, Y., Austin, E., Xiang, J., Gould, T., Larson, T., & Seto, E.: Health impact assessment of the 2020 Washington State wildfire smoke episode: Excess health burden attributable to increased PM2.5 exposures and potential exposure reductions. GeoHealth, 5, e2020GH000359. https://doi. org/10.1029/2020GH000359, 2021.*

   *Li, Y., Tong, D., Ma, S., Zhang, X., Kondragunta, S., Li, F., & Saylor, R.: Dominance of wildfires impact on air quality exceedances during the 2020 record-breaking wildfire season in the United States. Geophysical Research Letters, 48, e2021GL094908. https://doi.org/10.1029/2021GL094908, 2021.*

   We appreciate the useful information and comment. We have added them at Lines 57-59: "**Fires have a harmful effect on the climate and human society (Cascio., 2018; Li et al., 2021). On one hand, fires can substantially worsen air quality and endanger human health by spewing out harmful gases (Liu et al., 2021; O'Neill et al., 2021; Reid et al., 2016)**.".

2. *Introduction: Please include more papers on burned area trends, fire intensity trends, global patterns of wildfires, variations in fire season, and the impact of weather on wildfires.*

   Following the reviewer's suggestion, we added the introduction at Lines 77-91: "**In terms of BA, forest fires made up the majority of the area burned in Equatorial Asia, followed by the North America. Savanna fires were extremely prevalent in Africa and considerably less in South America. Farmland fires were the most prevalent in Europe and the Middle East, while grassland fires were dominant in Central Asia and South America (Giglio et al., 2013; van Wees et al., 2022). The majority of the world's regions, particularly those with forests in mid- and high-latitudes, will see a future with a higher danger of wildfires as global warming progresses (Yu and Ginoux, 2022; Zhu et al., 2021).**

   **In the past decade or so, although the reduction of man-made fires in tropical areas has led to the reduction of the global area of over-fire, the trend in other regions is on the rise, the frequency of extreme wildfire events is increasing,**

**and the difference in seasonal variation is more obvious (Bowman et al., 2020; Senande-Rivera et al., 2022; Zheng et al., 2021). Several recent studies (Huang et al., 2023; Xu et al., 2022) have found that the BA of wildfires in the West Bank of the United States and the Indo-China Peninsula in Southeast Asia has increased and has significant synoptic scale changes, and the strongest frequency spectrum is in the time scale of 1 week and 2 weeks, respectively. The former is controlled by wind speed and humidity, while the latter is mainly modulated by rainfall (Huang et al., 2023).**".

3. *Line 109: Please provide the full name for MODIS. Make sure you have provided the full name of any abbreviation before using it for the first time in your paper. Please check throughout the paper.*

   Sorry for the mistake. We have added the full spell for MODIS and checked the paper throughout.

4. *Line 110: full names for MCD, MCD64CMQ, and MCD14DL are missing.*

   We thank the reviewer's comments and have added their full names.

   According to the product guide, full name of MCD14DL is MODIS/Aqua+Terra Thermal Anomalies/Fire locations 1km V0061 NRT (Near real-time) distributed by LANCE FIRMS (https://www.earthdata.nasa.gov/learn/find-data/near-real-time/firms/mcd14dl-nrt). The combined (Terra and Aqua) MODIS NRT active fire products (MCD14DL) are processed using the standard MOD14/MYD14, where MOD14 (Terra) and MYD14 (Aqua) are Level 2 Fire Products. Therefore, MCD means a product that combines Aqua and Terra.

   MCD64 is the Collection 6 Moderate Resolution Imaging Spectroradiometer (MODIS) Burned Area product suite. The products with suffix CMQ refer to CMG fire products, where CMG refers to Climate Modeling Grid. Therefore, MCD64CMQ is the Climate Modeling Grid Burned Area Monthly Product (https://modis-fire.umd.edu/files/MODIS_C6_BA_User_Guide_1.3.pdf).

   Accordingly, we have added the descriptions in section 2.2.1 of the manuscript.

5. *Line 117: full names for MOD, and MYD are missing.*

   We have added the full names now.

6. *Line 117: Please add the reference for the FIRMS product here.*

   Added.

7. *Line 123: full name for FireCCI is missing.*

   Added.

8. *Section 2.2.2: Compare GFAS with other widely used fire emission products. (See general comments)*

   We have added the corresponding information as replied in the general comment.

Thanks.

9. *Line 134: full names for ERA, ERA-Interim, and ERA5 are missing.*
   Thanks for helping figure them out and we have added the full names now.

10. *Line 153: any reference for the definition of the fire month? Why do you use 80% rather than 70% or another percentage?*
    This is a good question. Here we refer to the study of Archibald et al. (2013). In their research, a monthly climatology of burned area was first produced from the entire time series. The monthly burned areas were then ranked, and the average fire season length was defined as the number of months required to reach 80% of the total average annual burned area (analysis of values from 70% to 85% showed that the results were not sensitive to the threshold chosen). Therefore, we have adopted 80% as the threshold in this study, and we have also added the reference at lines174-176: "**The month with 80% of the annual average burned area is the fire month, and the fire month number is the duration of the fire season (Archibald et al., 2013)**.".

    Archibald, S., Lehmann, C.E.R., Gómez-Dans, J.L., Bradstock, R.A.: Defining pyromes and global syndromes of fire regimes, Proceedings of the National Academy of Sciences, 110, 6442-6447, https://doi.org/10.1073/pnas.1211466110, 2013.

11. *Figure 1: It would be better to show the trend of burned areas in the 12 different regions defined in table 1.*
    The reviewer provides us useful comments. According to the current structure of this study, we have added the trend of burned areas in the 12 different regions as Figure R1 (Figure 4 in manuscript), and added the corresponding descriptions at lines 283-287: "**Although the global fire BA shows a downward trend, the changes in 12 regions are not consistent (Fig. 4). Specifically, the BA increased in WUS, SI, and SI regions, and decreased in NAF, SAF, and CE regions, while the trend change in other regions was not obvious. At the same time, we need point out that the regions with large BA are mainly located in low latitude, such as NAF, SAF and NAU, and their changes have a greater contribution to the reduction of global BA**.".

[Figure]

Figure R1. The trend of burned areas in the 12 different regions.

*12. Figure 1: Please explain the difference in the results of MODIS and FireCCI.*

We thank the reviewer's comments. After a careful inspection, we found that there were small errors in our previous programming, which led to differences between the two sets of data in Figure R2 (Figure 1 in manuscript). With the help of this comment, we have corrected all of them, and the results can still prove that the burning area shows a decreasing trend.

[Figure]

Figure R2. Global burned areas derived from MCD64 and FireCCI.

*13. Figure 2: Please add the time period in the figure 2 caption.*

Thanks, we have added it in Fig. 2 of the manuscript: "**Figure 2. The spatial distribution of global fire burned fraction (BF), fire count (FC) and fire radiation power (FRP) from 2001 to 2019.**".

*14. Line 199: It's interesting to see how land use affects the fire trend. It'd be better to add one figure and some discussion about the BA, BF, BC, and FRP trends in different land use type regions.*

We appreciate the reviewer's comments. We have another study dedicated to studying the spatiotemporal change characteristics of wildfire in different land use types, which is also under review. In order to avoid repetition of the topic content, we add a part of discussion at lines 223-227: "**From the perspective of land use type, the BA and FC show linear correlation in most cases. Specifically, in forests, fires mainly increase in temperate and boreal forest areas, and in farmland, fires mainly increase in South Asia and East Asia, while the area of global grassland fires generally shows a downward trend except for parts of the United States and Australia.**".

*15. Line 254: According to Sofiev et al. (2012), plume injection height is related to FRP. It'd be better to show the scatter plot of FRP and APT.*

*Sofiev, M., Ermakova, T., & Vankevich, R.: Evaluation of the smoke-injection height from wild-land fires using remote-sensing data. Atmospheric Chemistry and Physics, 12(4), 1995–2006. https://doi.org/10.5194/acp-12-1995-2012, 2012.*

This is a good suggestion. We carefully studied the above research and showed the relationship between plume injection height and FRP (Fig. R1). We find that Fig. R3 is consistent with the result (Figure R4) of Sofiev et al. (2012). It can be seen from the scatter plot that ATP is mainly distributed at 500-3500m and FRP is mainly distributed at 0-2000 MW, but there is obviously no significant linear relationship between the two indicators. In this study, we mainly want to analyze the relative changes of wildfire burning area, plume injection height and FRP in 12 regions to reflect the regional differences. Therefore, the relationship analysis between plume injection height and FRP is not included in this study. We need to further study the specific impact and response relationship between them in future.

[Figure]

Figure. R3. The scatter plot of plume injection height and FRP.

[Figure]

Figure R4. The scatter plot in the study of Sofiev et al. (2012).

16. *Line 264: The relative changes of FRP of NAU is not the highest. NCA, SI, and WUS is higher than NAU.*

We apologize for our wrong description. We have modified it at lines 299-300: "**The relative changes of BA, and APT of NAU are all the highest, and its FRP is also relatively high, as observed in Fig. 5, suggesting that the fire occurred frequently with high intensity**.".

17. *Line 274-303: This paragraph is very long. Consider separating it into two paragraphs.*

We appreciate this comment and have modified it as suggested.

18. *Figure 5: Can you explain why there is no data for SI in the Jan and Feb (2003-2011)?*

This is a good question. In principle, we are not sure about the reasons, while we are sure that the dataset did not provide the information in January and February from 2003 to 2011. One likely (we are not sure if it is) reason is the climate warming. We know that Siberia (SI) is in the high latitude of the northern hemisphere, where the winter is cold and long, making wildfires and their emissions rare or even nore in January and February at early stage such as before 2011. With the global warming, the wildfire becomes possible recently such as in February after 2011.

19. *Line 295: The increase in SI is only after May. There is a decrease from Jan-April.*

We have modified the sentence at lines 328-330: "**NCA and NEC showed a significant decrease in the mean value of emission fluxes from 2012 to 2020, while SI showed a significant increase in the mean value of emission fluxes from May to November during 2012-2020.**".

20. *Figure 6: Please provide the time period in the caption.*

Added.

*21. Line 388: why use 2001-2009 instead of 20 years (2001-2019)?*

Sorry for our typo and it is actually 2001-2019. We have corrected it at Lines 422-424: "**In 2001-2019, the precipitation in western Amazon decreased in summer, and the relative humidity and soil moisture showed a declining trend in summer and autumn, which were favorable for the occurrence and spread of fire**.".

---

## Author Comment (AC2)

We thank Dr. Xu for her thoughtful, valuable and detailed comments and suggestions that have helped us improve the paper quality. Our detailed responses (Blue) to Dr. Xu's questions and comments (*Italic*) are listed below.

*Reviewer #2:*

*This paper studied the patterns of global burned area, burned fraction, fire count, and fire intensity based on selections of remote sensing products. This paper also investigated the 12 fire-prone regions of their variations in fires, emissions, and how meteorological indicators of temperature, precipitation, relative humidity, and soil moisture impacted fires activities.*

We thank Dr. Xu for the detailed evaluations and valuable comments. We have gone through all the comments and revised the original manuscript based on the suggestions and comments.

*General comments:*
*1. My first concern is the statement of the research subject in this paper. This title and the writing of the paper were about wildfires while all the data used for this paper were not specifically just of wildfires. I need the authors to provide the definition of wildfires and evidence to prove the right use of the data (data for other fire types, e.g. prescribed fires, were removed in this analysis). Otherwise, the subject and the corresponding statements need to be changed to fires/biomass burning emissions.*

We thank the reviewer for the constructive suggestions. We have thought about this problem and carefully read the existing literature, and found that there is no reliable method to accurately distinguish fire and wildfire at a large spatial scale. We fully agree with the reviewer's idea and changed the relevant expression of the full manuscript to biomass burning emissions and change wildfires to fires. In addition, we still retain a few "wildfires" in the manuscript, which mainly follows the description in the corresponding references.

*2. I am also questioning the conclusion that wildfire burned area has decreased slowing over the last 20 years (line 17-18, 173). There was large difference of magnitude of the two burned area datasets and sometimes the trend was also different (year 2003-2005). The trend of FireCCI data is more stable than the MCD64 trend. Additionally, what does the data look like for the year 2020 and would it affect the results?*

These are good questions. The conclusion that the burned area has decreased slowly in the past 20 years in our study is credible, and the results in relevant studies are also consistent with our conclusions (e.g., Forkel et al., 2019; Zheng et al., 2021). However, there are differences in fire activities among different regions, and the burned area in some regions shows an increasing trend, such as the western United States and Siberia. To present the research results more clearly and accurately, we added the change trend of burned area in 12 regions as Figure 4 in the manuscript.

As for the difference between FireCCI and MCD64, we re-examined the data analysis and calculation process, and added the results of burned area in 2020. After careful inspection, we found some errors in our previous computer programming,

which led to big differences between the two sets of data in Figure R1. After correcting those errors, the results are consistent now which still prove that the burned area shows a decreasing trend.

Forkel, Matthias et al., Recent global and regional trends in burned area and their compensating environmental controls. Environ. Res. Commun. 1, 051005. https://doi.org/10.1088/2515-7620/ab25d2. 2019.

Zheng, B., Ciais, P., Chevallier, F., Chuvieco, E., Chen, Y., Yang, H. Increasing forest fire emissions despite the decline in global burned area, Sci. Adv., 7, eabh2646, https://www.science.org/doi/10.1126/sciadv.abh2646, 2021.

[Figure]

Figure R1. Global burned areas derived from MCD64 and FireCCI.

*3. I am concerned about the conclusion that "the increase in temperature in the northern hemisphere's middle and high latitude forest regions was primarily responsible for the increase in wildfires and emissions". This study has only looked at the four types of meteorology data and related them to fires. This conclusion could be misleading.*

We agree and thank the reviewer's comments. The meteorological variables selected in this study are indeed limited, and it is difficult to reflect all the causes of fire. However, the cause of fire is complex, even in terms of natural causes, the current research can hardly completely clarify it. So, our existing conclusions are based on the factor analysis results of this study on the one hand, and refer to the evidence in relevant studies (Engelmann et al., 2021; Jolly et al., 2015; Zhu et al., 2021) on the other hand.

Although the above studies also indicate that the increase of temperature leads to the increase of wildfires in the Arctic and Siberia, we have corrected the relevant description with a weak tone to be more reliable based on the suggestions of the reviewer at Lines 29-31: "**Correspondingly, the increase of temperature in the northern hemisphere's middle and high latitude forest regions is likely the major cause for the increase in fires and emissions, while the change in fires in tropical regions was largely influenced by the decrease in precipitation and relative humidity.**". In addition, we also revised the relevant description in the discussion

section.

Engelmann, R., Ansmann, A., Ohneiser, K., Griesche, H., Radenz, M., Hofer, J., Althausen, D., Dahlke, S., Maturilli, M., Veselovskii, I., Jimenez, C., Wiesen, R., Baars, H., Bühl, J., Gebauer, H., Haarig, M., Seifert, P., Wandinger, U., and Macke, A.: Wildfire smoke, Arctic haze, and aerosol effects on mixed-phase and cirrus clouds over the North Pole region during MOSAiC: an introduction, Atmos. Chem. Phys., 21, 13397–13423, https://doi.org/10.5194/acp-21-13397-2021, 2021.

Jolly, W., Cochrane, M., Freeborn, P. Holden, Z. A., Brown, T. J., Williamson G. J., and Bowman, D. M. J. S.: Climate-induced variations in global wildfire danger from 1979 to 2013, Nat Commun., 6, 7537, https://doi.org/10.1038/ncomms8537, 2015.

Zhu, X., Xu, X., and Jia, G.: Asymmetrical trends of burned area between eastern and western Siberia regulated by atmospheric oscillation, Geophys. Res. Lett., 48, e2021GL096095. https://doi.org/10.1029/2021GL096095, 2021.

*4. The third paragraph (line 59-70) in the introduction section could be reduced to one sentence and add to the first paragraph. This is one aspect of fire impacts and not a directly interest of this paper. The introduction part also lacks discussions about the current studies like this study, e.g., on global and regional fires and emissions. Some suggested readings:*

*Andela, N., Morton, D. C., Giglio, L., Chen, Y., van der Werf, G. R., Kasibhatla, P. S., ... & Randerson, J. T. (2017). A human-driven decline in global burned area. Science, 356(6345), 1356-1362.*

*Giglio, L., Randerson, J. T., & Van Der Werf, G. R. (2013). Analysis of daily, monthly, and annual burned area using the fourth‐generation global fire emissions database (GFED4). Journal of Geophysical Research: Biogeosciences, 118(1), 317-328.*

*van Wees, D., van der Werf, G. R., Randerson, J. T., Rogers, B. M., Chen, Y., Veraverbeke, S., ... & Morton, D. C. (2022). Global biomass burning fuel consumption and emissions at 500 m spatial resolution based on the Global Fire Emissions Database (GFED). Geoscientific Model Development, 15(22), 8411-8437.*

We thank the reviewer's valuable suggestions.

For the first point regarding the third paragraph, we carefully discussed it among the coauthors and think that it has important scientific value in this study. On the one hand, it shows the importance of fire and its emissions, and provides the latest research evidence. On the other hand, it proves the close relationship between fire and human society. Together with the description of the relationship between fire and natural environment in the second paragraph, they become the detailed discussion in the first paragraph of the introduction. Thus, we kept it in the revised version.

For the second point, we have made further discussions by following the suggestions of the reviewer at lines77-91: "**In terms of BA, forest fires made up the majority of the area burned in Equatorial Asia, followed by the North America. Savanna fires were extremely prevalent in Africa and considerably less in South America. Farmland fires were the most prevalent in Europe and the Middle East, while grassland fires were dominant in Central Asia and South America (Giglio et al., 2013; van Wees et al., 2022). The majority of the world's regions,**

**particularly those with forests in mid- and high-latitudes, will see a future with a higher danger of wildfires as global warming progresses (Yu and Ginoux, 2022; Zhu et al., 2021).**

**In the past decade or so, although the reduction of man-made fires in tropical areas has led to the reduction of the global area of over-fire, the trend in other regions is on the rise, the frequency of extreme wildfire events is increasing, and the difference in seasonal variation is more obvious (Bowman et al., 2020; Senande-Rivera et al., 2022; Zheng et al., 2021). Several recent studies (Huang et al., 2023; Xu et al., 2022) have found that the BA of wildfires in the West Bank of the United States and the Indo-China Peninsula in Southeast Asia has increased and has significant synoptic scale changes, and the strongest frequency spectrum is in the time scale of 1 week and 2 weeks, respectively. The former is controlled by wind speed and humidity, while the latter is mainly modulated by rainfall (Huang et al., 2023)**.".

*5. The methods and analysis of geographical detectors are not clear. How can the geographical detectors explain the percentage of the causes of fires?*

We apologize that we did not make it clear. In combination with the reviewer's suggestions in Specific comments, we have added the method description, equation and references of geographical detector at lines 182-193: "**We use geographic detector to quantify the contribution of meteorological conditions (temperature, relative humidity, soil moisture, and total precipitation) to fire changes in different regions. The geographic detector can explain the degree of variability of various independent variables (x) to dependent variable (y). The q statistic in the calculation results indicates the degree of interpretation of the corresponding variable and its value range is 0-1 (Eq.1). The larger the q is, the stronger the explanatory power of (x) to (y) is. The geographic detector model has currently been used extensively in research for quantitative attribution analysis (Wang et al., 2016; Zhang et al., 2019). Detailed description of this model can refer to the studies by Wang et al. (2010, 2016).**

$$q = 1 - \frac{\sum_{h=1}^{L} N_h \sigma_h^2}{N\sigma^2} \tag{1}$$

**where $h = 1, …, L$ is strata of y (burned area and intensity) or x (meteorological variable); $N_h$ and $N$ are the strata $h$ and the number of units in different fire regions; $\sigma_h^2$ and $\sigma^2$ are the variance of the strata $h$ and y value in the fire region respectively**.".

*Specific comments:*
*1. Line 21: "summer and autumn as the reasons with the most frequent wildfires worldwide". Is "reasons" a typo? "seasons"?*

Sorry for the typo and we have corrected it.

*2. line 24-26: "absolute amount of CO₂ produced by wildfires is the largest" is obvious according to the emission factors used in the emission models. I would recommend remove*

*this from the abstract.*

We agree with the reviewer and have removed it.

3. *Line 84-85: Why "debatable"? What are the observation inversion data?*

Sorry for the confusion. What we want to express here is that the fire emission information still has great uncertainty, and thus the research based on model simulation is likely more inaccurate than the research based on observation data. We modified the description at Lines 93-96: "**Meanwhile, the information regarding the emissions of various compounds caused by fires still has great uncertainty, and the model simulation results are likely more inaccurate than the observational data (Zhang et al., 2016; Zheng et al., 2021)**.".

4. *Line 90: "investigate the causes" is not appropriate.*

Corrected.

5. *Line 154: Why the month with 80% of the annual average burned area is the fire month?*

This is a good question.

Here we refer to the study of Archibald et al. (2013). In their research, a monthly climatology of burned area was first produced from the entire time series. The monthly burned areas were then ranked, and the average fire season length was defined as the number of months required to reach 80% of the total average annual burned area (analysis of values from 70% to 85% showed that the results were not sensitive to the threshold chosen). Therefore, we have adopted 80% as the threshold in this study, and have also added the reference at lines174-176: "**The month with 80% of the annual average burned area is the fire month, and the fire month number is the duration of the fire season (Archibald et al., 2013)**.".

Archibald, S., Lehmann, C.E.R., Gómez-Dans, J.L., Bradstock, R.A.: Defining pyromes and global syndromes of fire regimes, Proceedings of the National Academy of Sciences, 110, 6442-6447, https://doi.org/10.1073/pnas.1211466110, 2013.

6. *Line 164: I understand that Wang et al. (2010, 2016) provided a detailed explanation of q statistic. I would recommend adding a description of q statistic in this context for better reading experience.*

We appreciate this comment and have added the description as suggested.

7. *Line 171-174: Please add more explanation about the BA results and the difference from the two datasets.*

We thank the reviewer for the detailed suggestions. As replied in General comments, we have corrected Figure 1. Now, the burned area calculated by the two sets of data is consistent, and the burned area has shown a downward trend in the past 20 years.

8. *Line 255: How the relative changes calculated? Why using this index to investigate the*

*regional differences?*

This is a good question. Taking FEP as an example, *y* represents the relative changes, *a* represents absolute value of FRP in a region, *b* represents average value of FRP in 12 regions. The calculation process is shown in this equation: $y=(a-b)/b$. As shown above, this method can represent the relative change of the average state of a region compared with that of multiple regions. We have given the absolute values of fire area and emission change in different regions in the study (Figs. 4, 6 and 7), so we use the method of relative change to further explore the differences of 12 regions.

9. *8: how the annual trends were calculated? I didn't find the "*" in the figure represents that the trend has passed the 95% significance test.*

If we understand correctly, what the reviewer want to say here is Figure 8.

In this study, the trend analysis was carried out for the temperature, total precipitation, relative humidity and soil moisture at the global scale using the Mann–Kendall (M–K) τ test, with Sen's slope method. In this study, Sen's slope was applied to evaluate the strength of the trend value; then, the M–K statistical test was employed to test whether these estimated trends were significant at a given significance level. The relevant calculation process can refer to the research of Gui et al. (2021). We also added relevant descriptions at lines 169-172: "**In addition, the trend analysis was carried out for the climate data at the global scale using the Mann-Kendall (M-K) statistical test, with Sen's slope method. Specifically, Sen's slope was applied to evaluate the trend value; then, the M-K statistical test was employed to test whether these estimated trends were significant at a given significance level (Gui et al., 2021).**".

Yes, the grid points in the figure that passed the 95% significance test were indeed marked with the "*". Maybe it is not clear due to the limitation of color setting and picture frame, so we updated the Figure.

Gui, K., Che, H., Li, Lei, Zheng, Y., Zhang, L., Zhao, H., Zhong, J., Yao, W., Liang, Y., Wang, Y., Zhang, X.: The significant contribution of small-sized and spherical aerosol particles to the decreasing trend in total aerosol optical depth over land from 2003 to 2018. Engineering. https://doi.org/10.1016/j.eng.2021.05.017. 2021.

10. *Line 481: the link didn't work for me.*
Corrected.

11. *Figure S3-6: why only the period of 2001-2009?*

Sorry for the confusion. They are actually for the period of 2001-2019. In this study, the year in Figure S3-6 represents the starting year, that is, the sliding average result of 10-year from the starting year. In other words, taking 2001 in the figure as an example, the column of 2001 in the figure shows the result of sliding average from 2001 to 2010. We have added annotations in the figure and added specific descriptions in the title, as shown in the figure below.

[Figure]

Figure S3. The 10-year sliding average change trend of temperature in typical area. The "*" and "+" in the figure represent that the trend has passed the 95% and 90% significance tests. The year in the figure represents the starting year of the sliding average calculation.

---

## Author Response (AR2)

We thank the reviewers for their thoughtful, valuable and detailed comments and suggestions that have helped us improve the paper quality. Our detailed responses (Blue) to the reviewers' questions and comments (*Italic*) are listed below.

*Anonymous Referee #1*

*Overall, the manuscript has improved a lot compared to the first draft with more clarity and better organization. It will be a significant contribute to the field. I suggest "accepted as is".*

We highly appreciate the reviewer's invaluable suggests.

*Anonymous Referee #2*

*Thanks to the authors for their reply. I am glad to accept most their responses and revised manuscript. However, I do have a couple of concerns before recommending for publication:*

We appreciate the detailed comments from the reviewer which have helped us improve the paper quality.

*1. I would like to encourage the authors to make their code and data open to the public, or at least to the reviewers/editors. I appreciate that the authors honestly let us know that they found some errors in previous computer programming as for the difference between FireCCI and MCD64.*

We appreciate the reviewer's suggestion. The data used in this study were all downloaded and obtained from the official data website. We have already described the data source information in the manuscript at lines 518-525: "**The MOD14 product and FireCCI51 dataset can be downloaded from https://firms.modaps.eosdis.nasa.gov/ (last access: 25 February 2022), and https://climate.esa.int/en/projects/fire/data/ (last access: 12 February 2022) respectively. The MCD64CMQ product from https://modis-fire.umd.edu/ba.html (last access: 23 February 2023). Fire emission data from the Global Fire Assimilation System (GFAS) (https://apps.ecmwf.int/datasets/data/cams-gfas/, last access: 3 February 2022). ERA-5 Reanalysis data were provided by the European Centre for Medium Weather Forecasts, (https://cds.climate.copernicus.eu/, last access: 17 February 2022)."**.

Due to copyright considerations, it is not suitable for us to publish the data. Readers and researchers can obtain corresponding data from the above websites. In addition, the raw data occupies too much storage space, so we have uploaded the main data and code of this study to cloud storage (https://pan.baidu.com/s/1xGRjQtUIjzyS90Pbgg0LnQ, password: 8aml) for reviewing by reviewers and editors. The data for FireCCI, MCD64, and GFAS is in the "DATA" folder. The code includes the extraction and calculation of variables related to fire and its emissions, all in the "CODE" folder.

*2. I am still confused how q statistics works and how can the geographical detectors explain the percentage of the causes of fires. Open code and data would help to this question. On the other hand, my understanding of the q statistics citation (Wang et al. (2010, 2016)) is that this could be applied to measure the spatial stratified heterogeneity. Can the authors provide more explanation how this related to influence of meteorological factors on fires?*

Geodetector has an official website that introduces the principles of the model and specific operating steps (http://www.geodetector.cn/#_Download,_with_Datasets_1). I believe that most users obtain, learn, and use the software based on the detailed introduction on this official website. The software was developed using Excel 2007, R and QGIS, respectively. According to my understanding, users can choose a Geodetector software based on their actual situation. In this study, I chose the Excel version of the software, which does not require programming and can be run and output results according to the specific steps on the official website. So, there is no publicly

available code for this part of the calculation. We have uploaded the downloaded software package and the initial results after running the software to the "Geodetector" folder of cloud storage.

As for the second question, in my understanding, existing scientific researches have shown that natural fires are influenced by meteorological factors. Based on such prior knowledge, we conduct relevant analysis and explanations in our research. As for the spatial stratified heterogeneity, both the dependent variable (Y) and the independent variable (X) have spatial distribution, but it is not necessary to layer them in geographical space, and attribute stratification can also be done, that is, the stratification of Y or X can be in geographical space, time, or attribute. The above is based on my learning and understanding of this model. I believe that the introduction and explanation provided by the development team on the official website are more authoritative. We added the description at Lines 185-194: "**We use Geographical Detector (Geodetector) to quantify the contribution of meteorological conditions (temperature, relative humidity, soil moisture, and total precipitation) to fire changes in different regions. The Excel version of Geodetector used in this study was obtained from the development team's official website (http://www.geodetector.cn/#_Download,_with_Datasets_1). The Geodetector can explain the degree of variability of various independent variables (x) to dependent variable (y). Note that the layering of y or x can be geographical space, time, or attributes. The q statistic in the calculation results indicates the degree of interpretation of the corresponding variable and its value range is 0-1 (Eq.2). The larger the q is, the stronger the explanatory power of (x) to (y) is. The Geodetector has currently been used extensively in research for quantitative attribution analysis (Wang et al., 2016; Zhang et al., 2019). Detailed description of this model can refer to the studies by Wang et al. (2010, 2016**).".

*3. Same request to the relative change question (fig 4).*

The relative change is the calculation of the difference in the mean of each region relative to the average of 12 regions. This step is relatively simple, so I calculated them using the formula directly in the table, and the specific file have been uploaded to "the relative change" folder of the cloud storage. We added the description at Lines 180-184: "**In addition, the relative change is used in this study to represent the variation of fire characteristics in a region relative to all 12 regions (Eq.1).**

$$\text{RC} = (k_i - f_i)/f_i \tag{1}$$

**where RC represents the relative change, $k_i$ represents the mean of variable i in one region, and $f_i$ represents the mean of variable i in 12 study regions**.".

Minor:

*Line92: What does the "West Bank of the United States" mean? Do you mean "the Western United States"?*

Sorry for the confusion. We have corrected it to "**the West Coast of the United States**".

*Line 99-105: The authors stated here that "model simulation results are likely more inaccurate than the observational data". However, the data used from GFAS in this paper is also modeled data. This is confusing.*

We agree and thank the reviewer's comment. To avoid ambiguity, we modified the description are Lines 93-95: "**Meanwhile, the information regarding the emissions of various compounds caused by fires still has great uncertainty (Zhang et al., 2016; Zheng et al., 2021)**.". We chose GFAS in this study because it calculates biomass burning emissions by assimilating Fire Radiative Power (FRP) observations from the MODIS instruments onboard the Terra and Aqua satellites. It corrects for gaps in the observations, which are mostly due to cloud cover, and filters spurious FRP observations of volcanoes, gas flares and other industrial activity.

*Line 210-212: Please consider including the Mann-Kendall (M-K) statistical test and Sen's slope results if applicable. This suggestion applies to the rest of the analysis.*

We thank the reviewer for the comment. The Mann-Kendall (M-K) statistical test and Sen's slope method are important tools for geosciences spatial analysis, which are mainly used to identify and judge the trend changes and differences under the spatial distribution of research objects. It is not very suitable to only consider the relatively clear trend change analysis of fires at a time scale, such as shown in Fig. 1 and Fig. 4. Therefore, we have not conducted repeated analysis of this part for the time being, but thank the reviewer for the suggestion.

*Line 213-214, 231-234: Please rephrase the sentences like Line 249-250 when compared to other studies.*

We have corrected them.

*Line 340: Typo "Since" instead of "Science".*

Sorry for the typo and we have corrected it.

---

## Author Response (AR3)

We thank the reviewer for his/her thoughtful, valuable and detailed comments and suggestions that have helped us improve the paper quality. Our detailed responses (Blue) to the reviewer's questions and comments (*Italic*) are listed below.

*Anonymous Referee #2*

*Thanks to the authors for their positive reply. I suggest to accept once the authors clarified the following questions:*

We thank the reviewer for the detailed suggestions. We have gone through all the comments and revised the manuscript accordingly.

*1. Thanks for providing the data and code assess to reviewers. Sorry for the confusion about open data and code request. I didn't mean the source data used for the study, but the data generated by this project (the result data). I would strongly encourage the authors and editors to have a conversation to see if open source is possible at some degree.*

Follow the suggestions of the reviewer and editor, we added data related to the results at Lines 525-526: "**Data related to the results can be obtained from https://zenodo.org/record/7997467**.".

*2. There may be some misunderstanding here:*

*"Line 210-212: Please consider including the Mann-Kendall (M-K) statistical test and Sen's slope results if applicable. This suggestion applies to the rest of the analysis.*

*We thank the reviewer for the comment. The Mann-Kendall (M-K) statistical test and Sen's slope method are important tools for geosciences spatial analysis, which are mainly used to identify and judge the trend changes and differences under the spatial distribution of research objects. It is not very suitable to only consider the relatively clear trend change analysis of fires at a time scale, such as shown in Fig. 1 and Fig. 4. Therefore, we have not conducted repeated analysis of this part for the time being, but thank the reviewer for the suggestion."*

*In the paper the authors mentioned that "the trend analysis was carried out for the climate data at the global scale using the Mann-Kendall (M-K) statistical test, with Sen's slope method" (Revised version line 170-174). I didn't suggest using the two methods but intended to ask the authors to report results if applicable. Please let me know if I misunderstood anything there.*

Sorry for the confusion. We have stored the results of the M-K test and Sen's slope in the public link mentioned above.